# Structuring Semantic Embeddings for Principle Evaluation: A Kernel-Guided Contrastive Learning Approach

## Abstract

Reliable post-hoc evaluation asks whether already generated text satisfies a target criterion after generation. In this paper we study a focused frozen-embedding setting using principle-evaluation proxy tasks: toxicity detection, fine-grained emotion categorization, and ordinal review rating. General-purpose text embeddings are widely deployed for such tasks, but broad semantic similarity can place semantically similar yet task-distinct examples in overlapping regions of the representation space. We introduce Kernel-Guided Contrastive Learning (KGCL), a prototype-guided geometric regularization module built on top of frozen text embeddings. In KGCL, the objects historically called "kernels" are learnable class-level prototype anchors; they are not RKHS kernels, do not implement a kernel trick, and are not kernel-machine estimators. The module combines a semantic stream, a prototype-anchor attention stream, supervised contrastive learning, offset-based prototype-margin regularization, and stream regularization to produce a compact task-adapted representation without updating the base encoder. Controlled experiments show that KGCL improves over raw frozen embeddings on all three datasets and gives the clearest direct-baseline margin on AmazonReviews, while remaining competitive with strong direct frozen metric-learning baselines on GoEmotions and ToxicComment. We also add supervised residual-adapter, encoder-LoRA, full fine-tuning, objective ablation, sensitivity, and fully logged few-shot LLM protocol diagnostics to define the boundary of the claim. The theoretical analysis is revised as a sufficient-condition account for prototype-margin behavior under explicit assumptions in the prototype-mapping space, rather than as an unconditional training or final-embedding separation guarantee.

## 1 Introduction

Capturing task-relevant distinctions in text representations remains a fundamental challenge for post-hoc evaluation and natural language understanding (Weidinger et al., 2021; Bommasani et al., 2021; Hendrycks et al., 2023). While much of the alignment literature focuses on steering model behavior *during* text generation (Christiano et al., 2017; Ouyang et al., 2022; Bai et al., 2022), an equally important problem is how to determine whether a generated text satisfies a target criterion *after* it has been produced. This problem, commonly referred to as *post-hoc evaluation*, is central to applications such as content moderation, safety auditing, rating prediction, and large-scale monitoring of model outputs (Gehman et al., 2020; Rae et al., 2021). In this paper, we study a narrower setting than broad alignment evaluation: lightweight classification or rating tasks that serve as principle-evaluation proxies. The evaluated datasets cover toxicity detection, selected emotion categories, and ordinal review ratings; they are useful probes of representation quality, but they should not be interpreted as direct measurements of general safety, fairness, helpfulness, or human-value adherence.

At scale, post-hoc evaluation is typically built on high-dimensional text embeddings (Reimers & Gurevych, 2019; Neelakantan et al., 2022). Recent advances have produced large general-purpose embedding models, such as NV-Embed and GTE-style models, with billions of parameters (Li et al., 2023; Lee et al., 2024). Trained on diverse corpora, these models aim to learn broadly reusable semantic representations and achieve

---

**Input and frozen encoder:** text $x_i \rightarrow$ frozen encoder $E(\cdot) \rightarrow \mathbf{z}_i \in \mathbb{R}^{1024}$.

| | |
|---|---|
| **Semantic stream** | $\mathbf{z}_i \rightarrow \mathrm{MLP}_{sem} \rightarrow \mathbf{s}_i$ preserves task-relevant semantic context. |
| **Prototype stream** | $\mathbf{z}_i$, prototype anchors $\{\mathbf{c}_k\}_{k=1}^C \rightarrow$ query/key/value attention $\rightarrow \mathbf{m}_i$. |
| **Fusion** | $\tilde{\mathbf{e}}_i = \alpha\mathbf{s}_i + (1-\alpha)\mathbf{m}_i$, then $\mathbf{e}_i = \tilde{\mathbf{e}}_i/\|\tilde{\mathbf{e}}_i\|_2$. |
| **Losses** | supervised contrastive loss on $\mathbf{e}_i$; offset margin on $(\mathbf{m}_i, \mathbf{c}_{y_i})$; orthogonality between $\mathbf{s}_i$ and $\mathbf{m}_i$; optional ordinal magnitude loss on $\mathbf{m}_i$. |

---

Figure 1: Architecture and loss diagram for KGCL. The base text encoder is frozen. The learnable module contains a semantic MLP stream, a prototype-anchor attention stream, fusion and normalization, and the loss terms used for prototype-guided regularization. The prototype anchors are class-level reference points rather than RKHS kernels or kernel-machine estimators.

strong performance on broad benchmarks such as MTEB (Muennighoff et al., 2023). Consequently, they are widely used as frozen feature extractors for downstream evaluation tasks.

**Limitations of Existing Methods** Despite their strong general-purpose performance, these embedding models are not designed to resolve every fine-grained distinction required by a downstream evaluation task. Their objective of capturing broad semantic similarity can lead to representations that smooth over subtle but critical differences between texts. In many practical scenarios, texts can be highly similar at the lexical and semantic levels while reflecting different evaluation targets. For example, in e-commerce moderation, the reviews "*The package never arrived, terrible seller*" and "*The package never arrived, terrible courier*" share nearly identical surface semantics but correspond to distinct targets. Similarly, in content safety, a non-toxic expression such as "*I absolutely hate this garbage situation*" can be lexically similar to a toxic attack like "*I absolutely hate you, you are garbage*" (Gehman et al., 2020; Welbl et al., 2021). Standard embedding models—including those refined via unsupervised contrastive learning such as SimCSE (Gao et al., 2021)—can map such examples to overlapping regions of the representation space. We refer to this phenomenon as *representational entanglement*: semantically similar texts that correspond to different task labels are not cleanly organized according to the downstream label structure. This overlap can make frozen features less convenient for lightweight downstream probes and can obscure the subtle, context-dependent signals required for accurate post-hoc evaluation (Devlin et al., 2019; Zhou & Srikumar, 2022).

**Challenges for Tackling the Limitations** Addressing representational entanglement is challenging. Two common approaches are (i) adapting the embedding model via fine-tuning or parameter-efficient methods, or (ii) bypassing embeddings altogether by leveraging large language models (LLMs) through few-shot prompting (Qiu et al., 2020; Brown et al., 2020). Fine-tuning-based methods, including Parameter-Efficient Fine-Tuning (PEFT) techniques such as LoRA (Hu et al., 2022), can be effective when raw text and training compute are available, and we report separate adaptation diagnostics in the revised experiments. They nevertheless change the deployment setting relative to frozen-embedding post-hoc evaluation. On the other hand, leveraging in-context learning (ICL) with large LLMs for evaluation introduces nontrivial protocol, latency, and cost considerations (Dong et al., 2024). Using generative models for discriminative tasks can be difficult to deploy at scale (Zheng et al., 2023; Wang et al., 2023). Taken together, these considerations motivate lightweight methods that can improve a frozen embedding store through a small task-specific module, while keeping the base encoder fixed.

**Proposed Method** To address this gap, we propose *Kernel-Guided Contrastive Learning (KGCL)*, a lightweight and plug-and-play framework for task-adapting frozen text embeddings for post-hoc evaluation proxy tasks. KGCL operates on top of frozen base embeddings and projects them into a low-dimensional, task-specific subspace through an attention-based transformation. In this subspace, we introduce a set of learnable *prototype anchors* that serve as class-level reference points for the evaluated labels. The term "kernel" is retained as the method name but should be read as a historical naming convention, not as RKHS kernel learning. To reduce label-relevant entanglement in the frozen-feature setting, we further design a geometry-aware contrastive objective augmented with an *offset penalty*. This objective encourages texts associated with the same label to cluster near their corresponding anchors, while increasing the relative

margin from competing anchors. By adding task-relevant geometric regularization to fixed representations, KGCL produces task-adapted embeddings for lightweight post-hoc evaluation.

**Contributions** Our contributions are threefold. First, we propose a lightweight and modular architecture that maps general-purpose text embeddings into a structured subspace for principle-evaluation proxy tasks, without updating the underlying encoder. Second, we introduce a prototype-guided training objective that combines supervised contrastive learning, offset-based prototype margins, and stream regularization; our theory is presented as a sufficient-condition analysis of the geometry encouraged by this objective. Third, we add controlled comparisons against direct metric-learning baselines (SupCon projection head, center loss, triplet loss, prototype classifier, ArcFace, and CosFace), supervised adaptation diagnostics, objective ablations and sensitivity analyses, and a fully logged local few-shot LLM protocol. The revised evidence supports a bounded empirical claim: KGCL improves over raw embeddings across the evaluated proxy tasks and provides a favorable same-protocol comparison against direct frozen metric-learning baselines, with its clearest margin on AmazonReviews and smaller margins on GoEmotions and ToxicComment; encoder-updating adaptation methods remain a distinct setting.

## 2 Related Work

**General-Purpose Text Embeddings and Subspace Learning.** Massive general-purpose embedding models have become the foundational paradigm for text representation. However, while these models excel at capturing broad semantic context, they can conflate subtle task-specific distinctions (Devlin et al., 2019; Zhou & Srikumar, 2022). To extract task-relevant features from these entangled spaces, prior works have explored subspace learning and task-specific projections (Fukumizu et al., 2003; Edelman & Intrator, 1997; Peng et al., 2019). Yet, general dimensionality reduction techniques, such as UMAP (McInnes et al., 2018) or Spectral Embedding (Von Luxburg, 2007), are primarily optimized for structural visualization rather than supervised task-specific representation learning. Similarly, while geometric embeddings successfully structure representation spaces for tasks like knowledge graph querying (Ren et al., 2020), they are not designed for the frozen text-embedding post-hoc evaluation setting studied here. Our framework addresses this by learning a structured, low-dimensional subspace on top of fixed embeddings and evaluating it through downstream classifiers.

**Metric Learning and Prototype-Based Classification.** KGCL is closely related to supervised metric learning rather than being an unrelated alternative. Contrastive and supervised contrastive objectives encourage same-label examples to cluster while separating different labels (Hadsell et al., 2006; Chen et al., 2020; Khosla et al., 2020). Large-margin metric learning and triplet-style objectives impose relative distance constraints (Weinberger & Saul, 2009; Schroff et al., 2015). Center loss and prototypical methods introduce class-level anchors or prototypes (Wen et al., 2016; Snell et al., 2017), while angular-margin classifiers such as CosFace and ArcFace improve separation on normalized representations (Wang et al., 2018; Deng et al., 2019). KGCL belongs to this broader family. Its specific contribution is to study a frozen-text-embedding setting with a dual stream projection, prototype-anchor attention, offset-based prototype margins, and stream regularization. For this reason, the revised experiments compare against SupCon projection heads, center loss, triplet loss, prototype classifiers, ArcFace, and CosFace under the same frozen-embedding protocol.

**Parameter-Efficient Fine-Tuning (PEFT).** To adapt massive pre-trained models for specific downstream tasks, Parameter-Efficient Fine-Tuning (PEFT) methods, such as Low-Rank Adaptation (LoRA) (Hu et al., 2022), have been widely adopted to mitigate the prohibitive costs of full fine-tuning. These methods can be strong when raw text, labels, and training compute are available. They nevertheless update the encoder or attach trainable modules inside the encoder computation, whereas KGCL studies a post-hoc setting in which embeddings are already generated and the base encoder remains fixed. Because these are different compute settings, the revised experiments report supervised residual adapters, encoder-LoRA, and full fine-tuning as separate adaptation diagnostics rather than merging them into the direct frozen-embedding baseline table.

**Principle Alignment and LLM-Based Evaluation.** Prior efforts in principle alignment predominantly focus on constraining language models *during* the text generation process, utilizing techniques such as Reinforcement Learning from Human Feedback (RLHF) (Christiano et al., 2017; Ouyang et al., 2022) and Constitutional AI (Bai et al., 2022). However, evaluating already generated text remains a distinct post-hoc problem. Recently, a prevailing trend has been to leverage the In-Context Learning capabilities of massive LLMs to act as evaluators (Zheng et al., 2023). Such comparisons are highly protocol-dependent: prompt format, demonstration selection, decoding settings, invalid-output handling, latency, and cost can materially change the outcome (Wang et al., 2023; Dong et al., 2024). We therefore treat the LLM comparison as a specified few-shot protocol rather than as a general claim that KGCL outperforms all LLM evaluators.

## 3 Methodology

### 3.1 Problem Formulation

Let us define the pre-trained embedding space as a metric space $(\mathbb{R}^D, d_{sem})$, where a frozen general-purpose encoder $E(\cdot)$ maps textual inputs from a dataset $\mathcal{X} = \{x_i\}_{i=1}^N$ into high-dimensional vectors $\mathbf{z}_i \in \mathbb{R}^D$. Given a discrete label space $\mathcal{Y} = \{y_1, \ldots, y_C\}$ for a post-hoc classification or rating task, the bottleneck studied here is *feature entanglement*. Because the distance metric $d_{sem}$ is optimized for broad semantic similarity, it can remain agnostic to the task-specific boundaries of $\mathcal{Y}$. Consequently, two text instances $x_i$ and $x_j$ may share similar lexical structure while requiring different labels ($y_i \neq y_j$), yet still occupy nearby regions in the frozen embedding space. Such overlap can reduce the usefulness of frozen features for lightweight downstream evaluators.

To reduce this entanglement without updating the base encoder, we formulate the task as learning a lightweight mapping function $f_\theta : \mathbb{R}^D \to \mathcal{S}^{d-1}$ (where $d \ll D$ and $\mathcal{S}^{d-1}$ denotes the hypersphere manifold) alongside a set of learnable prototype anchors $\mathcal{K} = \{\mathbf{c}_1, \ldots, \mathbf{c}_C\} \in \mathcal{S}^{d-1}$. The term "kernel" in KGCL is a naming convention for these prototype anchors; it does not refer to RKHS kernels, a kernel trick, or classical kernel-machine learning. The optimization encourages the following margin pattern in the projected subspace:

$$\forall i, j: \quad ||f_\theta(\mathbf{z}_i) - \mathbf{c}_{y_i}||_2 \leq \delta_{intra}, \quad ||\mathbf{c}_{y_i} - \mathbf{c}_{y_j}||_2 \geq \delta_{inter} \tag{1}$$

where $\delta_{intra}$ is the target intra-class compactness radius and $\delta_{inter}$ is the target inter-anchor separation margin. For ordinal regression tasks, this categorical prototype-margin target is extended with an ordinal bias in the prototype-mapping stream. The unit-norm angular initialization arranges prototype anchors in an ordered starting configuration, while the magnitude regularizer acts on the pre-normalized prototype mapping $\mathbf{m}_i$. Thus ordinal intensity is encouraged before the final hyperspherical normalization rather than inferred from the norm of the final embedding $\mathbf{e}_i$. These constraints should be read as objective-level regularizers and sufficient-condition targets, not as unconditional guarantees that every training run will realize the ideal margins.

### 3.2 Kernel-Guided Principle Extractor

The primary objective of the neural principle extractor, $f_\theta$, is to project the frozen semantic vectors $\mathbf{z}_i \in \mathbb{R}^D$ onto a low-dimensional task-specific manifold $\mathcal{S}^{d-1}$. However, directly mapping inputs to label-specific coordinates risks *representation collapse*, where text instances lose their lexical diversity and merge into indistinguishable points.

To circumvent this, we design a *Dual-Stream Architecture* governed by specific inductive biases. We hypothesize that a useful task-adapted representation should balance two information flows: (1) a structural context that preserves the necessary semantic background, and (2) a dynamic projection that aligns the input with label-level prototype anchors.

**Semantic Basis Stream (Contextual Regularization).** To reduce the risk of representation collapse, the first stream provides a semantic baseline. We project the original embedding $\mathbf{z}_i$ through a shared Multi-Layer Perceptron (MLP):

$$\mathbf{s}_i = \text{MLP}_{sem}(\mathbf{z}_i) \in \mathbb{R}^d$$

where $\mathbf{s}_i$ denotes the semantic basis. This pathway acts as an information bottleneck that preserves lexical nuances (e.g., the specific object being reviewed) independent of the targeted principle, ensuring the resulting metric space remains continuous. Detailed specifications for this MLP are provided in Appendix A.1.

**Prototype Mapping Stream (Dynamic Manifold Projection).**   The second stream is the core mechanism for reducing feature entanglement. Instead of relying only on hyperplanes for linear classification, we introduce a matrix of learnable prototype anchors, $\mathbf{C} = [\mathbf{c}_1, \mathbf{c}_2, \ldots, \mathbf{c}_C]^\top \in \mathbb{R}^{C \times d}$. These anchors serve as explicit, optimizable class-level reference points for the $C$ labels, providing coordinate centers for margin-based distance calculations.

To determine the input's alignment with each principle, we employ an attention mechanism as a soft projection operator. We project both the input $\mathbf{z}_i$ and the prototype matrix $\mathbf{C}$ into a joint space using linear weights $\mathbf{W}_q \in \mathbb{R}^{D \times d}$ and $\mathbf{W}_k, \mathbf{W}_v \in \mathbb{R}^{d \times d}$:

$$\mathbf{q}_i = \mathbf{z}_i \mathbf{W}_q, \quad \mathbf{K} = \mathbf{C}\mathbf{W}_k, \quad \mathbf{V} = \mathbf{C}\mathbf{W}_v$$

The attention distribution $\mathbf{a}_i \in \mathbb{R}^C$ computes the coordinate coefficients of the input query against each principle's basis:

$$\mathbf{a}_i = \text{Softmax}\left(\frac{\mathbf{q}_i \mathbf{K}^\top}{\sqrt{d}}\right)$$

The principle-specific mapping $\mathbf{m}_i$ is then dynamically constructed as a convex combination of the projected prototype values:

$$\mathbf{m}_i = \mathbf{a}_i \mathbf{V} \in \mathbb{R}^d$$

Task-specific initialization strategies for these prototype anchors are detailed in Appendix A.2.

**Feature Fusion and Manifold Projection.**   Finally, the extractor fuses the contextual semantics ($\mathbf{s}_i$) with the principle-specific alignment ($\mathbf{m}_i$). We then project the fused vector onto the hypersphere manifold used by the geometric regularizers

$$\tilde{\mathbf{e}}_i = \alpha \mathbf{s}_i + (1-\alpha)\mathbf{m}_i, \qquad \mathbf{e}_i = \frac{\tilde{\mathbf{e}}_i}{||\tilde{\mathbf{e}}_i||_2} \in \mathcal{S}^{d-1}$$

where $\alpha \in [0, 1]$ is a learnable gating parameter initialized to a small scalar. This normalization places the final representation $\mathbf{e}_i$ on the target hypersphere and prepares it for the subsequent geometry-aware contrastive optimization.

### 3.3   Geometry-Aware Contrastive Objective

While the dual-stream extractor $f_\theta$ (Section 3.2) provides the structural capacity to project representations onto the hypersphere $\mathcal{S}^{d-1}$, this architecture alone does not determine a useful task geometry. Without explicit regularization, the prototype anchors $\mathbf{C}$ may remain arbitrary vectors, the attention mechanism may become uninformative, and the semantic stream $\mathbf{s}_i$ may redundantly encode task-specific signals. To activate the inductive biases designed in $f_\theta$, we train the network with complementary geometric regularizers.

Relying solely on standard contrastive learning is not always sufficient for this purpose. Traditional InfoNCE objectives optimize relative probabilities via softmax, which encourages general clustering but does not directly control distances to class-level anchors. Consequently, semantically overlapping classes can still suffer from boundary entanglement. We therefore construct a composite objective $\mathcal{L}_{\text{total}}$ in which each component plays a distinct regularizing role tailored to the dual-stream architecture. Specifically, this composite objective integrates four complementary mechanisms: a *Supervised Contrastive Loss* to encourage coarse class-level clustering, an *Offset Loss* to encourage prototype-margin behavior, an *Orthogonality Loss* to reduce redundancy between semantic context and prototype mapping, and an optional *Magnitude Loss* to preserve ordinal intensity progressions.

**Global Clustering via Supervised Contrastive Loss ($\mathcal{L}_{\textbf{contrastive}}$).** Before learning fine-grained task boundaries, the model benefits from a coarse task-level topology. We utilize a Supervised InfoNCE loss to encourage initial task-specific clusters. By utilizing target labels, it pulls the fused representation $\mathbf{e}_i$ towards positive samples $\mathbf{e}_{p_i}$ sharing the same principle $y_i$, while broadly repelling samples from differing principles. This supplies an initial macroscopic bias in the metric space.

**Prototype-Margin Regularization via Offset Loss ($\mathcal{L}_{\textbf{offset}}$).** To complement the relative separation provided by contrastive learning with an explicit anchor-based margin, we introduce the Offset Penalty. This acts as our primary geometric regularizer. It controls the coordinates of the prototype mapping $\mathbf{m}_i$ relative to the prototype anchors and encourages the margin pattern formulated in Eq. 1. It operates through two tandem constraints:

- **Intra-Class Penalty (Bounding Variance):** To discourage clusters from expanding indefinitely and to keep semantic diversity within a local neighborhood, we introduce a "safe radius" $\delta_{\text{intra}}$. It penalizes samples only if they drift beyond this distance from their target prototype $\mathbf{c}_{y_i}$:

$$P_{\text{intra},i} = \max(0, ||\mathbf{m}_i - \mathbf{c}_{y_i}||_2 - \delta_{\text{intra}})^2$$

- **Inter-Class Penalty (Encouraging Separation):** To reduce feature entanglement, this penalty encourages a sample to be closer to its true prototype than to any incorrect prototype $\mathbf{c}_k$ by a target margin $\delta_{\text{inter}}$:

$$P_{\text{inter},i} = \max(0, ||\mathbf{m}_i - \mathbf{c}_{y_i}||_2 - \min_{k \neq y_i} ||\mathbf{m}_i - \mathbf{c}_k||_2 + \delta_{\text{inter}})^2$$

The batch-level offset loss is the dynamically weighted expectation of these penalties, where $w_{y_i}$ serves to counteract class imbalance:

$$\mathcal{L}_{\text{offset}} = \frac{1}{B} \sum_{i=1}^{B} w_{y_i}(\lambda_{\text{inclass}} P_{\text{intra},i} + \lambda_{\text{crossclass}} P_{\text{inter},i})$$

**Information Decoupling via Orthogonality Loss ($\mathcal{L}_{\textbf{orthogonality}}$).** For the dual-stream architecture (Section 3.2) to function properly, the Semantic Basis ($\mathbf{s}_i$) and the Prototype Mapping ($\mathbf{m}_i$) should avoid redundant encoding. If $\mathbf{s}_i$ absorbs all label-specific signals, the prototype-anchor stream contributes little. We therefore apply a soft Orthogonality Loss that penalizes high cosine similarity between the two streams:

$$\mathcal{L}_{\text{orthogonality}} = \frac{1}{B} \sum_{i=1}^{B} w_{y_i} \max(0, |\cos(\mathbf{s}_i, \mathbf{m}_i)| - \delta_{\text{orthogonal}})$$

**Structural Progression via Magnitude Loss ($\mathcal{L}_{\textbf{magnitude}}$).** When the evaluation principles exhibit an inherent ordinal relationship (e.g., 1 to 5 star ratings), treating them as independent categorical clusters ignores the severity of misclassification (e.g., confusing 1-star with 5-star is worse than with 2-star). KGCL uses two complementary ordinal biases. First, ordinal prototype anchors are initialized as an angular progression on the unit hypersphere, giving adjacent ratings closer starting anchors than distant ratings. Second, the Magnitude Loss acts on the pre-normalized prototype mapping $\mathbf{m}_i$ and encourages its scale to follow the label intensity $I(y_i)$:

$$\mathcal{L}_{\text{magnitude}} = \frac{1}{B} \sum_{i=1}^{B} w_{y_i}(||\mathbf{m}_i||_2 - \lambda_{\text{scale}} I(y_i) \cdot ||\mathbf{c}_{y_i}||_2)^2$$

Because the final representation $\mathbf{e}_i$ is normalized, ordinal intensity should be understood as a training-time bias in the prototype-mapping/fusion process rather than as a claim that final embedding norms encode rating intensity.

**Overall Objective.** The final network is trained end-to-end by minimizing the weighted summation:

$$\mathcal{L}_{\text{total}} = \lambda_{\text{offset}}\mathcal{L}_{\text{offset}} + \lambda_{\text{contrastive}}\mathcal{L}_{\text{contrastive}} + \lambda_{\text{orth}}\mathcal{L}_{\text{orthogonality}} + \lambda_{\text{mag}}\mathcal{L}_{\text{magnitude}}$$

The magnitude term, $\lambda_{\text{mag}}\mathcal{L}_{\text{magnitude}}$, is activated exclusively for ordinal tasks. The training configurations, including optimizer setups and hyperparameter selection, are provided in Appendix A.3. A computational complexity analysis of this objective is presented in Appendix A.4.

### 3.4 Theoretical Bounds on Geometric Quality

A core motivation for explicitly designing the Offset Loss ($\mathcal{L}_{\text{offset}}$) is to bias the projected subspace ($\mathbb{R}^d, d_{\text{task}}$) toward compact within-class neighborhoods and separated class-level anchors. In this section, we state a sufficient-condition analysis: if the learned prototype mapping satisfies bounded within-class spread and sufficient inter-anchor margin, then conditional margin and clustering-quality statements follow. Detailed mathematical proofs and extended discussions on empirical error bounds and architecture justifications are provided in Appendix B.

**Theorem 1 (Conditional Prototype-Margin Bound).** Let $\mathcal{M}_A$ and $\mathcal{M}_B$ denote the sets of prototype-mapped representations belonging to two distinct labels $A$ and $B$, with prototype anchors $\mathbf{c}_A$ and $\mathbf{c}_B$. Suppose examples remain within $\delta_{\text{intra}} + \varepsilon$ of their corresponding prototype anchor and the two prototype anchors are separated by at least $\delta_{\text{inter}}$. If $\delta_{\text{inter}} > 2\delta_{\text{intra}} + 2\varepsilon$, then the Euclidean distance between any $\mathbf{m}_a \in \mathcal{M}_A$ and $\mathbf{m}_b \in \mathcal{M}_B$ is lower-bounded by a positive constant:

$$||\mathbf{m}_a - \mathbf{m}_b||_2 > 0 \tag{2}$$

**Theorem 2 (Conditional Bounds on Geometric Clustering Metrics).** Building upon the same assumptions and given that the effective margins satisfy $\delta_{\text{inter}} > 4\delta_{\text{intra}} + 2\varepsilon$, the prototype-mapping space admits an upper bound for the Within/Between distance ratio and a positive lower bound for the Silhouette Score:

$$\text{Ratio}_{W/B} \leq \frac{2\delta_{\text{intra}}}{\delta_{\text{inter}} - 2\delta_{\text{intra}} - 2\varepsilon} \qquad \text{and} \qquad S(\mathbf{m}_i) > 0 \tag{3}$$

**Discussion.** These theorems state sufficient mathematical requirements for a structured prototype-mapping space. *Theorem 1* states that inter-prototype margins, bounded intra-class deviation, and small optimization error imply a non-trivial lower bound on inter-class distances in the prototype-mapping stream. *Theorem 2* connects these constraints to standard clustering metrics, indicating that a sufficiently large inter-anchor margin is sufficient for a positive Silhouette Score ($S > 0$). These statements do not assert that gradient-based training automatically reaches the assumptions, nor that every final fused normalized embedding directly inherits the same margins; the final embeddings are evaluated empirically in Section 4.4.

**Implications.** The practical value of these bounds lies in clarifying the geometric bias of the objective. If the sufficient conditions approximately hold, the projected representation may have lower within-class variation and more controlled inter-class margins. This can reduce the empirical burden on lightweight linear probes. For ordinal settings, the framework also accommodates monotonic intensity progressions as a training-time bias. The empirical sections below test these effects through downstream weighted F1, AUC, macro F1, ordinal diagnostics, and geometry diagnostics rather than treating the sufficient-condition analysis as a stand-alone guarantee.

## 4 Experiment

In this section, we empirically evaluate the downstream behavior, efficiency, and geometry diagnostics of the Kernel-Guided Contrastive Learning (KGCL) framework under a frozen-embedding post-hoc protocol. Our evaluation is designed to answer four questions: (1) Does KGCL improve downstream linear-probe performance and empirical geometry over raw frozen embeddings? (2) How does it compare with direct metric-learning and prototype baselines under the same frozen-embedding protocol? (3) How sensitive is

the composite objective to its components and weights? (4) How does the reported few-shot LLM baseline behave under a fully specified protocol?

## 4.1 Experimental Setup

**Datasets.** To test our framework on semantically overlapping texts, we select three challenging datasets that serve as principle-evaluation proxy tasks rather than broad alignment benchmarks: **(1) GoEmotions (Demszky et al., 2020):** A large-scale corpus of Reddit comments. We focus on a confusable subset of five emotion labels (Disappointment, Sadness, Disapproval, Gratitude, Approval). Because these emotions frequently share similar vocabulary (e.g., dense negative sentiment), this task tests whether the learned representation improves downstream discrimination in an entangled subjective-label space. **(2) Amazon Reviews (Ni et al., 2019):** Comprising user reviews and 1-5 star ratings, this dataset acts as a proxy for ordinal review-intensity evaluation. We use it to evaluate whether the learned geometry supports both categorical distinctions and ordinal progression. **(3) Toxic Comment Classification Challenge (cjadams et al., 2017):** This dataset evaluates toxicity detection under strong class imbalance. It presents lexical conflation cases where non-toxic venting can resemble toxic attacks. We utilize an extremely unbalanced test set (approximately 1:25 toxic vs. non-toxic), mirroring real-world moderation scenarios. The training set is resampled to a 1:3 ratio to facilitate stable learning.

**Base Encoder and Baselines.** Across all primary frozen-embedding experiments, we utilize `jina-embeddings-v3` (Sturua et al., 2024) ($D = 1024$) as our frozen base encoder, owing to its strong performance in capturing broad semantic similarity. Our neural principle extractor projects these into a $d = 64$ dimensional subspace (dimension justification in Appendix A.5). We compare against four tiers of baselines: (1) **Raw Embeddings:** Direct evaluation on the frozen `jina-embeddings-v3` features using linear probes; (2) **Unsupervised Contrastive Learning:** General structure-enhancing methods like SimCSE (Gao et al., 2021); (3) **Direct Metric-Learning and Prototype Baselines:** supervised contrastive projection head, center loss, triplet loss, prototype classifier, ArcFace, and CosFace, all trained on the same frozen embeddings; and (4) **Protocol-Specific LLM and Adaptation Diagnostics:** a fully logged local few-shot LLM protocol, a supervised residual adapter on frozen embeddings, and separate encoder-updating LoRA/full fine-tuning diagnostics.

**Evaluation Metrics.** Due to class imbalance in datasets like Toxic Comment and Amazon Reviews, we emphasize weighted F1, macro F1, AUC, precision, and recall where applicable. The original downstream classifier tables are reported as Mean $\pm$ Standard Deviation over 10-fold cross-validation. The newly added direct metric-learning baselines use the same frozen-embedding and linear-probe protocol; a two-seed diagnostic is used to identify the strongest direct baseline per dataset. To empirically diagnose representation geometry, we compute standard geometric indices alongside our composite *Geometric Quality Index (GQI)*. The GQI combines inter-class margin, intra-class compactness, silhouette behavior, and local class overlap; a higher GQI is interpreted as an empirical geometry signal for downstream probes, not as a formal separation proof.

**Split and Leakage-Control Protocol.** Because KGCL uses labels during representation learning, the revised experiments use an explicit train/validation/test separation for the recovered embedding arrays. KGCL training uses train labels only; checkpoint, representation, and threshold choices use validation labels only; downstream probes are fit on the training split; test labels are used only once for final metric computation. A split audit verified that the expected arrays and label distributions are present and that test-label information is not used in representation learning or model selection.

Table 1: Fixed split sizes used in the revised leakage-control audit.

| Dataset | Train | Validation | Test |
|---|---|---|---|
| GoEmotions | 5126 | 906 | 881 |
| AmazonReviews | 2271 | 487 | 487 |
| ToxicComment | 44256 | 15073 | 22609 |

## 4.2 Core Task Validation: Downstream Probe Performance

To evaluate our framework, we first isolate the representational gain achieved by the KGCL module. We compare the optimized embeddings against raw embeddings to test whether the prototype-guided objective improves downstream probe performance and empirical geometry.

**Downstream Probe Performance (GoEmotions).** A critical question is whether the performance improvements stem from a more useful task-adapted representation or merely from adding a downstream classification head. To address this, we use a controlled "Simple Fine-Tuning Baseline" on the GoEmotions five-principle subset. We freeze the base embeddings and train an identical suite of classifiers (SVM, Random Forest, Logistic Regression, XGBoost, Transformer) on both the high-dimensional raw embeddings (1024-d) and our optimized representations (64-d).

As summarized in Table 2, training standard classifiers directly on raw embeddings plateaus at an Overall F1-score around 0.72-0.73. However, replacing the input with our optimized embeddings yields improvements across the reported metrics in this diagnostic. For instance, Logistic Regression (LR) F1 improves from 0.726 $\pm$ 0.031 to 0.776 $\pm$ 0.032. This improvement supports the practical utility of prototype-guided geometry for downstream probes; it should be read as an empirical probe result rather than a theorem-level final-embedding separation claim.

Table 2: Overall (Avg. Principle) Performance on GoEmotions Five-Principle Set (Mean $\pm$ Std. Dev.)

| Metric | Emb. Type | SVM | RF | LR | XGBoost | Transformer |
|---|---|---|---|---|---|---|
| Precision | Raw Emb. | $0.748 \pm 0.049$ | $0.733 \pm 0.059$ | $0.737 \pm 0.031$ | $0.747 \pm 0.020$ | $0.785 \pm 0.031$ |
|  | Opt. Emb. | $\mathbf{0.787} \pm 0.035$ | $\mathbf{0.789} \pm 0.036$ | $\mathbf{0.791} \pm 0.033$ | $\mathbf{0.773} \pm 0.028$ | $\mathbf{0.787} \pm 0.029$ |
| Recall | Raw Emb. | $0.721 \pm 0.045$ | $0.737 \pm 0.031$ | $0.722 \pm 0.031$ | $0.741 \pm 0.014$ | $0.763 \pm 0.024$ |
|  | Opt. Emb. | $\mathbf{0.764} \pm 0.034$ | $\mathbf{0.769} \pm 0.039$ | $\mathbf{0.772} \pm 0.033$ | $\mathbf{0.765} \pm 0.039$ | $\mathbf{0.769} \pm 0.031$ |
| F1 | Raw Emb. | $0.729 \pm 0.046$ | $0.722 \pm 0.035$ | $0.726 \pm 0.031$ | $0.737 \pm 0.018$ | $0.764 \pm 0.036$ |
|  | Opt. Emb. | $\mathbf{0.770} \pm 0.033$ | $\mathbf{0.767} \pm 0.036$ | $\mathbf{0.776} \pm 0.032$ | $\mathbf{0.764} \pm 0.026$ | $\mathbf{0.770} \pm 0.030$ |

**Extension to Ordinal Intensities (Amazon Reviews).** As an ordinal extension of prototype-guided regularization, we evaluate the framework on rating intensities (1-5 star ratings). As shown in Table 3, optimized embeddings improve overall regression metrics (e.g., MSE, RMSE) compared to raw embeddings, suggesting that the structured subspace can support continuous ordinal constraints together with categorical distinctions.

Table 3: Overall Ordinal Regression Performance on Amazon Reviews (Mean $\pm$ Std. Dev.)

| Metric | Emb. Type | SVM | RF | LR | XGBoost | Transformer |
|---|---|---|---|---|---|---|
| MSE | Raw Emb. | $0.668 \pm 0.135$ | $0.506 \pm 0.120$ | $0.635 \pm 0.097$ | $0.546 \pm 0.163$ | $0.602 \pm 0.173$ |
|  | Opt. Emb. | $\mathbf{0.365} \pm 0.158$ | $\mathbf{0.392} \pm 0.143$ | $\mathbf{0.394} \pm 0.149$ | $\mathbf{0.377} \pm 0.097$ | $\mathbf{0.359} \pm 0.086$ |
| RMSE | Raw Emb. | $0.813 \pm 0.083$ | $0.706 \pm 0.083$ | $0.795 \pm 0.059$ | $0.731 \pm 0.111$ | $0.768 \pm 0.110$ |
|  | Opt. Emb. | $\mathbf{0.593} \pm 0.119$ | $\mathbf{0.617} \pm 0.107$ | $\mathbf{0.618} \pm 0.112$ | $\mathbf{0.609} \pm 0.080$ | $\mathbf{0.595} \pm 0.071$ |
| $R^2$ | Raw Emb. | $0.604 \pm 0.086$ | $0.700 \pm 0.074$ | $0.624 \pm 0.060$ | $0.677 \pm 0.095$ | $0.643 \pm 0.103$ |
|  | Opt. Emb. | $\mathbf{0.785} \pm 0.089$ | $\mathbf{0.770} \pm 0.080$ | $\mathbf{0.768} \pm 0.083$ | $\mathbf{0.777} \pm 0.055$ | $\mathbf{0.788} \pm 0.047$ |

To make the ordinal claim more direct, we also compute ordinal classification diagnostics on the validation-selected AmazonReviews representation. Table 4 shows that KGCL reduces not only weighted classification error but also rating-distance error. Compared with raw embeddings and the strongest direct SupCon baseline, KGCL lowers MAE, increases quadratic weighted kappa (QWK), and reduces severe errors, defined as predictions more than one rating level away from the label.

**Minority-Class Behavior Under Imbalance (Toxic Comments).** For evaluation in a sensitive domain, we assess our framework on the highly unbalanced Toxic Comment dataset. Table 5 shows that

Table 4: AmazonReviews ordinal diagnostics on the held-out test split.

| Method | Weighted F1 | MAE ($\downarrow$) | QWK ($\uparrow$) | Severe error ($\downarrow$) |
|---|---|---|---|---|
| Raw embeddings | 0.589 | 0.4784 | 0.8175 | 0.0637 |
| SupCon projection | 0.628 | 0.4271 | 0.8319 | 0.0472 |
| KGCL | **0.727** | **0.3039** | **0.8904** | **0.0205** |

optimized embeddings improve both Average F1 and Minority F1 across most classifiers. This suggests that the learned representation can help lightweight classifiers recover minority-class signal under strong imbalance, although the later direct-baseline comparison shows that center loss is also highly competitive on this task.

Table 5: Performance on Toxic Comment Classification Challenge (Mean $\pm$ Std. Dev.)

| Metric | Emb. Type | SVM | RF | LR | XGBoost | Transformer |
|---|---|---|---|---|---|---|
| Avg. F1 | Raw Emb. | $0.932 \pm 0.004$ | $0.949 \pm 0.003$ | $0.897 \pm 0.004$ | $0.918 \pm 0.004$ | $0.956 \pm 0.004$ |
| | Opt. Emb. | $\textbf{0.938} \pm 0.004$ | $\textbf{0.949} \pm 0.004$ | $\textbf{0.936} \pm 0.003$ | $\textbf{0.943} \pm 0.004$ | $\textbf{0.959} \pm 0.004$ |
| Minority F1 | Raw Emb. | $0.497 \pm 0.025$ | $0.405 \pm 0.044$ | $0.396 \pm 0.018$ | $0.433 \pm 0.024$ | $0.574 \pm 0.027$ |
| | Opt. Emb. | $\textbf{0.507} \pm 0.024$ | $\textbf{0.537} \pm 0.035$ | $\textbf{0.493} \pm 0.023$ | $\textbf{0.518} \pm 0.028$ | $\textbf{0.589} \pm 0.023$ |

### 4.3 Paradigm Comparisons: Direct Baselines and LLMs

After reporting the improvement over raw embeddings, we benchmark KGCL against alternative learning paradigms. The central additional comparison is against direct metric-learning and prototype baselines trained under the same frozen-embedding protocol.

**Direct Metric-Learning and Prototype Baselines.** Table 6 reports the direct baselines requested by the reviewers. All methods start from the same frozen `jina-embeddings-v3` features and are evaluated with the same downstream linear-probe weighted F1 protocol. KGCL has the largest direct-baseline margin on AmazonReviews. On GoEmotions and ToxicComment, the margins are small: the best direct baseline is close to KGCL on GoEmotions and center loss is highly competitive on ToxicComment. The strongest two-seed direct-baseline means are 0.620 on AmazonReviews (SupCon), 0.760 on GoEmotions (center loss), and 0.952 on ToxicComment (center loss), while KGCL reaches 0.953 on ToxicComment. We therefore report KGCL as a favorable same-protocol frozen-embedding comparison, not as a universal metric-learning superiority claim.

Table 6: Direct metric-learning and prototype baselines under the same frozen-embedding linear-probe protocol. Values are test weighted F1.

| Dataset | RAW | KGCL | SupCon | Center | Triplet | Prototype | ArcFace | CosFace |
|---|---|---|---|---|---|---|---|---|
| AmazonReviews | 0.589 | **0.727** | 0.628 | 0.597 | 0.592 | 0.599 | 0.588 | 0.566 |
| GoEmotions | 0.726 | **0.772** | 0.757 | 0.755 | 0.759 | 0.764 | 0.747 | 0.752 |
| ToxicComment | 0.915 | **0.953** | 0.949 | 0.951 | 0.943 | 0.947 | 0.948 | 0.949 |

The same predictions are also evaluated with paired bootstrap confidence intervals. These intervals support positive KGCL differences over the strongest direct baseline on AmazonReviews and GoEmotions, while the ToxicComment center-loss comparison remains a small competitive margin whose confidence interval overlaps zero.

**Separate Adaptation Diagnostics.** Table 8 reports supervised adaptation diagnostics separately from the direct frozen-embedding table. A supervised residual adapter on frozen embeddings obtains 0.614/0.786/0.925 on AmazonReviews/GoEmotions/ToxicComment. KGCL improves over the frozen residual adapter on AmazonReviews and ToxicComment, while GoEmotions favors the adapter in this diagnostic. Encoder-LoRA and full encoder fine-tuning update an MPNet encoder and obtain higher scores on

Table 7: Paired bootstrap differences in weighted F1. Positive CIs support a positive KGCL difference; ToxicComment is reported as a small competitive margin.

| Dataset | Comparison | Difference | 95% CI |
|---|---|---|---|
| AmazonReviews | KGCL − best direct SupCon | 0.0904 | [0.0443, 0.1377] |
| GoEmotions | KGCL − best direct center loss | 0.0256 | [0.0011, 0.0473] |
| ToxicComment | KGCL − best direct center loss | 0.0009 | [-0.0012, 0.0029] |

GoEmotions and ToxicComment. These results define the boundary of the claim: KGCL is a lightweight frozen-embedding post-hoc method, not a replacement for PEFT or full supervised fine-tuning.

Table 8: Separate supervised adaptation diagnostics. Values are test weighted F1. Encoder-LoRA and full fine-tuning update an MPNet encoder and are not part of the frozen-embedding direct-baseline table.

| Dataset | KGCL ref. | Frozen residual adapter | Encoder-LoRA | Full fine-tuning |
|---|---|---|---|---|
| AmazonReviews | **0.727** | 0.614 | 0.573 | 0.589 |
| GoEmotions | 0.772 | 0.786 | **0.831** | 0.820 |
| ToxicComment | 0.953 | 0.925 | 0.959 | **0.965** |

**Overcoming the Limitations of Task-Agnostic Contrastive Learning.** We also compare KGCL against the structure-enhancing contrastive baselines used in the original submission. As detailed in Table 9, maximizing general semantic similarity via Unsupervised SimCSE degrades downstream performance compared to raw embeddings (Average F1 dropping from 0.620 to 0.531). This illustrates how task-agnostic optimization can conflate crucial task boundaries (e.g., confusing "logistics failure" with "product defect" due to shared negative sentiment). Conversely, KGCL achieves a large improvement over SimCSE and remains higher than the standard supervised baseline in this AmazonReviews diagnostic.

Table 9: Downstream Classification Performance on Amazon Reviews: Contrastive Baselines Comparison (AUC / Overall F1-Score)

| Embedding | SVM | RF | LR | XGB | Trans. | **Avg. F1** |
|---|---|---|---|---|---|---|
| Unsupervised SimCSE | 0.851 / 0.568 | 0.823 / 0.539 | 0.843 / 0.572 | 0.840 / 0.567 | 0.730 / 0.407 | 0.531 |
| Standard Supervised | 0.921 / 0.697 | 0.916 / 0.718 | 0.918 / 0.687 | 0.922 / 0.687 | 0.921 / 0.687 | 0.695 |
| **KGCL (Ours)** | **0.934 / 0.709** | **0.933 / 0.749** | **0.928 / 0.719** | **0.930 / 0.724** | **0.927 / 0.726** | **0.725** |

**Comparison with Few-shot Large Language Models.** To make the LLM comparison reproducible, we treat it as a specified few-shot protocol rather than as a general LLM-evaluator claim. The completed telemetry uses a locally hosted `qwen3.6:27b` model through Ollama with deterministic decoding (`temperature=0.0`, `top_p=1.0`, `max_tokens=8`). The appendix reports prompt structure, demonstration selection, parsing rules, invalid-output handling, call counts, token counts, latency, and cost status.

Table 10 summarizes this audited run. Under this local-Qwen protocol, KGCL obtains higher weighted F1 on all three evaluated tasks. The monetary API cost is $0.00 because the run is local; hardware and energy cost are not monetized. We do not claim general superiority over all LLM evaluators.

## 4.4 Geometric Analysis, Ablation, and Efficiency

To understand the mechanics behind KGCL's empirical behavior, we analyze the geometric properties of the learned subspace, ablate its core loss components, and evaluate its practical deployment advantages.

**Quantitative Geometric Analysis and the GQI Metric.** To empirically diagnose the geometry discussed in Section 3.4, we evaluate the representation space using standard geometric metrics: the ratio of Within-class to Between-class Variance (Fisher, 1936), Silhouette Score (Rousseeuw, 1987), and Class Overlap (Dom, 2012). Furthermore, to provide a holistic empirical summary of the subspace's geometry, we

Table 10: Audited local `qwen3.6:27b` few-shot LLM protocol telemetry.

| Task | Calls | Invalid rate | Avg latency | P95 latency | LLM F1 | KGCL F1 |
|------|-------|--------------|-------------|-------------|--------|---------|
| ToxicComment | 520/520 | 0.000 | 1.182s | 1.310s | 0.940 | 0.947 |
| GoEmotions | 881/881 | 0.000 | 0.793s | 0.859s | 0.680 | 0.772 |
| AmazonReviews | 500/500 | 0.000 | 1.310s | 1.576s | 0.648 | 0.727 |

formulate the *Geometric Quality Index (GQI)*:

$$GQI = \left(1 - \frac{\text{Within Variance}}{\text{Between Variance}}\right) \times \text{Silhouette Score} \times (1 - \text{Class Overlap}) \tag{4}$$

Class Overlap is computed as the mean fraction of each sample's 10 nearest neighbors that carry a different label. A higher GQI indicates a space where clusters are internally compact and externally well-separated. We use GQI only as a reproducible geometry diagnostic; it is not treated as an empirical proof of the sufficient-condition theorem.

As detailed in Table 11 for the Amazon Reviews dataset, raw embeddings and task-agnostic methods like SimCSE do not yield favorable geometry diagnostics. Notably, SimCSE exhibits a Within/Between Ratio greater than 1.0 (indicating intra-class variance exceeds inter-class distance), resulting in a negative GQI (-0.0005). KGCL obtains a lower Within/Between Ratio of 0.358 and a higher GQI of 0.0975. These diagnostics support the geometric motivation of the method, but they are not treated as proof that every final fused embedding satisfies the sufficient-condition assumptions.

Table 11: Quantitative Geometric Quality and GQI Comparison on Amazon Reviews

| Embedding | W/B ratio ($\downarrow$) | Silhouette ($\uparrow$) | Class overlap ($\downarrow$) | **GQI ($\uparrow$)** |
|-----------|--------------------------|-------------------------|------------------------------|----------------------|
| Raw Embeddings | 8.760 | 0.018 | 0.563 | $< 0.000$ |
| Unsupervised SimCSE | 1.010 | 0.010 | 0.491 | -0.0005 |
| Standard Supervised | 0.458 | 0.158 | 0.286 | 0.0614 |
| **KGCL (Ours)** | **0.358** | **0.203** | **0.253** | **0.0975** |

The revised all-dataset diagnostic is reported in Appendix D.4. AmazonReviews and GoEmotions show improved GQI relative to raw embeddings and the strongest direct baseline. ToxicComment behaves differently because strong class imbalance makes the Within/Between component unstable; it is therefore reported as a diagnostic rather than as an advantage claim.

**Qualitative Visualization.** This quantitative pattern is consistent with the t-SNE (Van der Maaten & Hinton, 2008) visualizations (Figure 2). The raw embeddings (Figures 2a and 2c) show representational overlap among semantically related labels. After applying KGCL (Figures 2b and 2d), the embeddings become more organized around task-specific regions. For ordinal tasks (Amazon), the clusters also exhibit a visible ordered progression. These plots are qualitative diagnostics and are interpreted together with the quantitative tables rather than as standalone evidence of guaranteed separation.

**Objective Ablation and Sensitivity.** To isolate the contribution of each training objective, the original submission reported a GoEmotions ablation study (Table 12). The revised experiments add same-provenance retraining and loss-weight diagnostics across all three datasets. The resulting picture is task-dependent: geometric regularization is useful, but the full composite objective is not uniformly best under every retraining configuration. We therefore present the objective as a configurable prototype-guided regularization design rather than as a universally necessary set of loss terms.

**Architecture-Level Diagnostics.** To separate architecture behavior from loss-term behavior, we further evaluate stream and prototype variants requested by the reviewers. Table 15 reports weighted F1. On GoEmotions, the prototype-attention stream is useful: attention-only reaches 0.7718, while semantic/no-attention variants reach 0.7371. On ToxicComment, attention alone is insufficient, but full and no-attention/fixed

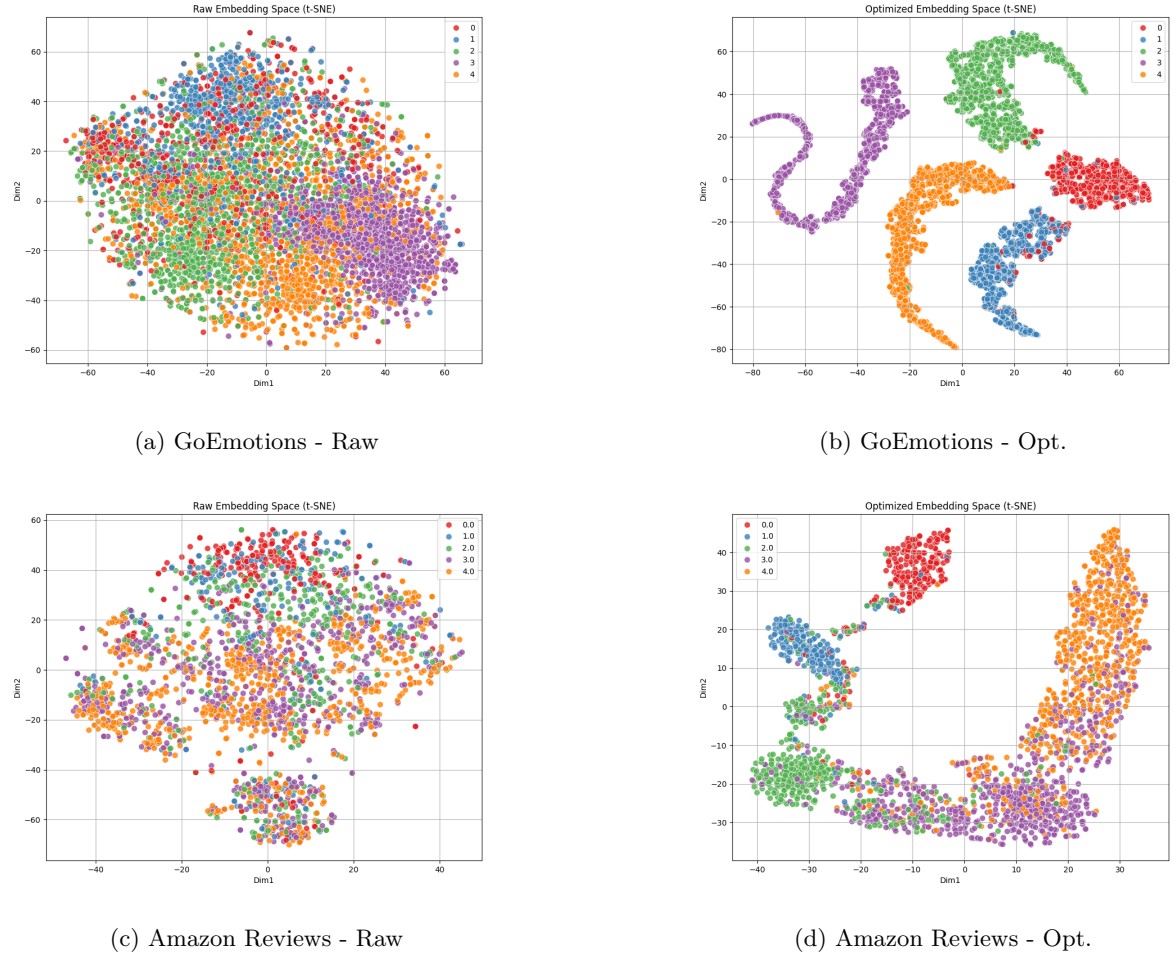

Figure 2: t-SNE comparison of embedding spaces. Raw embeddings (a, c) show overlap among related labels. KGCL optimized embeddings (b, d) show more organized task-specific structure and a visible ordinal pattern for Amazon Reviews.

Table 12: Ablation study on GoEmotions (F1 score Mean). Disappt.-Disappointment, Sad.-Sadness, Disapprv.-Disapproval, Grat.-Gratitude, Apprv.-Approval.

| Configuration | Disappt. | Sad. | Disapprv. | Grat. | Apprv. | **Average** |
|---|---|---|---|---|---|---|
| Only Contrastive Loss | 0.37 | 0.75 | 0.72 | 0.92 | 0.74 | 0.75 |
| Only Offset Loss | 0.40 | 0.75 | 0.72 | 0.94 | 0.77 | 0.77 |
| Without Contrastive Loss | 0.42 | 0.77 | 0.72 | 0.94 | 0.77 | 0.77 |
| Without Offset Loss | 0.44 | 0.71 | 0.73 | 0.93 | 0.76 | 0.77 |
| Raw Embeddings | 0.36 | 0.64 | 0.66 | 0.93 | 0.72 | 0.72 |
| **KGCL (Full Model)** | **0.49** | **0.77** | **0.72** | **0.94** | **0.76** | **0.78** |

variants remain close, indicating that the imbalanced task is less diagnostic for every component. Amazon-Reviews retrained architecture ablations are mixed, so we additionally use the submitted-checkpoint stream diagnostic: features, enhanced, and normalized enhanced representations all remain around 0.719-0.724, while attention-only is weak at 0.4603. This is consistent with the learned mapping/fusion representation without claiming that any single component universally drives performance.

Table 13: Same-provenance objective ablation across datasets. Values are test weighted F1.

| Dataset | Full | w/o offset | w/o contrast. | w/o orth. | Magnitude |
|---------|------|------------|---------------|-----------|-----------|
| AmazonReviews | 0.569 | 0.541 | **0.643** | 0.571 | 0.577 |
| GoEmotions | 0.748 | 0.753 | 0.748 | **0.756** | N/A |
| ToxicComment | 0.944 | 0.936 | 0.944 | **0.946** | N/A |

Table 14: Objective sensitivity and interaction diagnostics. Values are test weighted F1.

| Dataset | Full | $w_o$=0.5 | $w_o$=2.0 | $w_r$=1.0 | $w_r$=10.0 | Best interaction |
|---------|------|-----------|-----------|-----------|------------|------------------|
| AmazonReviews | 0.584 | 0.525 | **0.664** | 0.578 | 0.577 | Class+offset: 0.637 |
| GoEmotions | 0.745 | 0.757 | **0.758** | 0.756 | 0.744 | Class+contrastive(+offset): 0.754 |
| ToxicComment | 0.945 | 0.941 | 0.947 | **0.949** | 0.945 | Class+contrastive+offset: 0.946 |

**Beyond Accuracy: Deployment Efficiency.** Ultimately, the value of learning a task-adapted embedding extends beyond metric gains. By mapping the 1024-dimensional raw vectors into a 64-dimensional task-specific subspace, KGCL provides a reusable intermediate representation for the frozen-embedding post-hoc pipeline. This dimensionality reduction can reduce downstream training and inference cost for lightweight classifiers. For instance, the training and inference times for downstream models like XGBoost were reduced by up to 96.5% compared to using raw embeddings. This efficiency claim is specific to the frozen-embedding pipeline and is separate from encoder-updating adaptation methods.

## 5 Conclusion and Future Work

In this paper, we address feature entanglement in frozen text embeddings for post-hoc principle-evaluation proxy tasks. Rather than updating the base encoder, KGCL learns a lightweight prototype-guided mapping that regularizes frozen embeddings into a compact task-specific representation. The theory is stated as a sufficient-condition analysis for the prototype-mapping space: if within-class spread is bounded, inter-prototype separation is large enough, and optimization residuals remain small, then margin and clustering-quality consequences follow. The final fused normalized embeddings are evaluated empirically rather than treated as directly covered by a theorem.

The revised experiments support a bounded empirical claim. KGCL improves over raw frozen embeddings on GoEmotions, AmazonReviews, and ToxicComment. Against direct frozen metric-learning baselines, it has the largest positive margin on AmazonReviews, a smaller positive margin on GoEmotions, and a competitive result with center loss on ToxicComment, where the bootstrap confidence interval for the center-loss comparison overlaps zero. Separate encoder-updating diagnostics show that LoRA and full fine-tuning can obtain higher scores than KGCL on GoEmotions and ToxicComment. KGCL should therefore be understood as a frozen-embedding post-hoc method, not as a replacement for supervised encoder adaptation. The LLM comparison is similarly protocol-specific: under the audited local `qwen3.6:27b` few-shot protocol, KGCL is higher on the evaluated tasks, but broader LLM-as-a-judge comparisons remain future work.

Several limitations define the next stage of the work. The evaluated datasets are proxies for emotion distinctions, ordinal review intensity, and toxicity detection; they do not by themselves serve as general alignment, fairness, helpfulness, or human-value evaluation benchmarks. The objective ablations show task-dependent component effects, so future work should study validation-driven objective selection and broader multi-seed checks. Multimodal extensions to CLIP or VLM embeddings are plausible but not yet validated.

The content-moderation use case also raises broader-impact concerns. False negatives can miss harmful content, while false positives can suppress non-toxic speech, especially for minority dialects, non-standard expressions, or protected-attribute language. Prototype geometry can inherit and concentrate training-label bias, and a high-throughput evaluator can apply a flawed principle definition at scale. KGCL should therefore

Table 15: Architecture-level diagnostics. Values are weighted F1 means across available seeds unless noted.

| Dataset | Full ref. | Attn. only | Sem./no-attn. | Fixed proto. | Fixed $\alpha$ | w/o orth. |
|---|---|---|---|---|---|---|
| AmazonReviews | 0.6217 | 0.5073 | 0.6225 | 0.6170 | 0.6253 | 0.6148 |
| GoEmotions | 0.7519 | **0.7718** | 0.7371 | 0.7544 | 0.7481 | 0.7507 |
| ToxicComment | 0.9494 | 0.9146 | 0.9483 | 0.9469 | 0.9472 | **0.9500** |

be used with human oversight, threshold calibration, auditability, and mechanisms for contesting moderation decisions.

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

# A    Implementation Details

This appendix provides further details regarding the neural principle extractor's architecture, prototype-anchor initialization strategies, specific training hyperparameters and loss function configurations, computational complexity, and justification for the task-specific subspace dimension, as referenced in the main paper.

## A.1    Neural Network Architecture Details

The neural principle extractor $f_\theta$ is implemented as a neural network that maps the input text embedding $\mathbf{X}_i \in \mathbb{R}^{1024}$ to a $d = 64$ dimensional principle-aware representation $\mathbf{e}_i$. The architecture is composed of a shared Multi-Layer Perceptron (MLP) and an attention mechanism.

The shared MLP used to compute the semantic basis $\mathbf{s}_i$ consists of two fully connected layers with LeakyReLU activation functions and Batch Normalization. Dropout is applied after each hidden layer for regularization. The layer dimensions are as follows:

- Input layer: $\mathbb{R}^{1024} \to \mathbb{R}^{512}$

- Hidden layer 1: $\mathbb{R}^{512} \to \mathbb{R}^{256}$ (followed by LeakyReLU, Batch Norm, Dropout)

- Hidden layer 2: $\mathbb{R}^{256} \to \mathbb{R}^{d}$ (followed by LeakyReLU, Batch Norm, Dropout), where $d = 64$. The output of this layer is the semantic basis $\mathbf{s}_i$.

The Dropout rate used throughout the MLP is 0.2.

The attention mechanism involves linear transformations of the input embedding and the prototype anchors to compute queries, keys, and values:

- Query projection: query_fc : $\mathbb{R}^{1024} \to \mathbb{R}^{d}$

- Key projection: key_fc : $\mathbb{R}^{d} \to \mathbb{R}^{d}$

- Value projection: value_fc : $\mathbb{R}^{d} \to \mathbb{R}^{d}$

These projected vectors are used in the scaled dot-product attention calculation.

The learnable parameter $\alpha$ that weights the semantic basis and prototype-anchor mapping in the final fusion layer is a scalar variable initialized to 0.05.

## A.2    Prototype-Anchor Initialization Details

The $K$ learnable prototype anchors $\mathbf{c}_k \in \mathbb{R}^{d}$ are initialized based on the task type to encourage structured learning. These anchors were called kernels in the original method naming, but they are not RKHS kernels or kernel-machine estimators.

**For Classification Tasks:**    For classification tasks (GoEmotions five-emotion subset, Amazon Reviews classification, and ToxicComment), the $K$ prototype anchors are initialized randomly on the unit hypersphere in $\mathbb{R}^{d}$. To encourage distinct starting points, we apply a procedure targeting a minimum pairwise Euclidean distance between any two initialized anchors. This initialization does not guarantee final orthogonality or final separation. A target minimum distance of $\sqrt{2}$ (the Euclidean distance between orthogonal unit vectors) is aimed for during this initialization step.

**For Ordinal Regression Tasks:**    For ordinal ratings (e.g., 1 to 5 stars), the prototype anchors are initialized on the unit hypersphere with a sequential angular progression along a defined geodesic path. This provides an ordinal inductive bias: distant ratings are initialized farther apart than adjacent ratings. The initialization is a bias rather than a proof that every learned representation preserves a strict global ordering.

### A.3 Training and Loss Function Details

This appendix provides the full mathematical formulations for each component of the composite objective $\mathcal{L}_{\text{total}}$ referenced in Section 3.3, along with specific training configurations.

The neural principle extractor is trained end-to-end using the AdamW optimizer with an initial learning rate of 1e-4 and a weight decay of 1e-5. A learning rate scheduler (Cosine Annealing or ReduceLROnPlateau) dynamically adjusts the learning rate based on validation performance. Training is capped at 100 epochs with early stopping to prevent overfitting, using a consistent batch size of 128. To counteract class imbalance, dynamic class weights $w_{y_i}$ are applied across all loss calculations, computed as the inverse frequency of each true class within the current training batch.

**Contrastive Loss ($\mathcal{L}_{\textbf{contrastive}}$).**  To encourage macroscopic class-level clustering, we employ the standard Supervised Contrastive Loss (SupCon), which uses label information to pull together samples from the same principle while repelling negatives:

$$\mathcal{L}_{\text{contrastive}} = \frac{1}{B} \sum_{i=1}^{B} \frac{-w_{y_i}}{|P(i)|} \sum_{p \in P(i)} \log \frac{\exp(\text{sim}(e_i, e_p)/\tau)}{\sum_{a \in A(i)} \exp(\text{sim}(e_i, e_a)/\tau)} \tag{5}$$

where $A(i)$ is the set of all other samples in the batch excluding $i$, $P(i)$ is the set of positive samples sharing the same label $y_i$ as anchor $i$, $|P(i)|$ is its cardinality, $\text{sim}(\cdot, \cdot)$ denotes cosine similarity, and $\tau = 0.1$ is the temperature scaling factor.

**Offset Loss ($\mathcal{L}_{\textbf{offset}}$).**  The total offset loss combines the intra-class and inter-class penalties:

$$\mathcal{L}_{\text{offset}} = \frac{1}{B} \sum_{i=1}^{B} w_{y_i} \left( \lambda_{\text{inclass}} P_{\text{intra},i} + \lambda_{\text{crossclass}} P_{\text{inter},i} \right)$$

where $\lambda_{\text{inclass}}$ and $\lambda_{\text{crossclass}}$ dictate the relative strengths of the penalties, and the margins $\delta_{\text{intra}}$ and $\delta_{\text{inter}}$ typically range within $[0.1, 0.5]$.

**Orthogonality Loss ($\mathcal{L}_{\textbf{orthogonality}}$).**  Promotes "soft" orthogonality between the semantic basis $\mathbf{s}_i$ and prototype mapping $\mathbf{m}_i$:

$$\mathcal{L}_{\text{orthogonality}} = \frac{1}{B} \sum_{i=1}^{B} w_{y_i} \cdot \max(0, |\cos(\mathbf{s}_i, \mathbf{m}_i)| - \delta_{\text{orthogonal}})$$

where the dynamic margin $\delta_{\text{orthogonal}}$ is linearly annealed from 0.5 down to 0.05 during training.

**Magnitude Loss ($\mathcal{L}_{\textbf{magnitude}}$).**  Applied exclusively for ordinal regression tasks to enforce natural ordering:

$$\mathcal{L}_{\text{magnitude}} = \frac{1}{B} \sum_{i=1}^{B} w_{y_i} \left( ||\mathbf{m}_i||_2 - \lambda_{\text{scale}} I(y_i) \cdot ||\mathbf{c}_{y_i}||_2 \right)^2$$

where $\lambda_{\text{scale}}$ is a learnable scaling factor and $I(y_i)$ maps the label to a numerical intensity (e.g., $I(y_i) = y_i$).

The final total objective ($\mathcal{L}_{\text{total}}$) is optimized by balancing these components.

### A.4 Computational Complexity Analysis

We analyze the computational complexity of our framework during training and inference.

**Training Complexity:**  The primary computational cost during training arises from the forward and backward passes through the neural principle extractor and the calculation of the loss components over a batch of size $B$. The extractor involves:

- Shared MLP: A sequence of matrix multiplications. Given input dimension $D = 1024$, output dimension $d = 64$, and hidden dimensions $h_1 = 512, h_2 = 256$, the complexity is $O(D \cdot h_1 + h_1 \cdot h_2 + h_2 \cdot d)$ per sample.

- Attention Mechanism: Involves linear projections ($O(D \cdot d + d^2 \cdot K)$ for a batch of size $B$, where $K$ is the number of principles), computing attention scores ($O(B \cdot K \cdot d)$), and weighted summation ($O(B \cdot K \cdot d)$).

The dominant part of the forward pass per batch is approximately $O(B \cdot (D \cdot h_1 + h_1 \cdot h_2 + h_2 \cdot d + K \cdot d))$. Loss calculations involve vector operations and distance calculations on the $d$-dimensional embeddings and $K$ prototype anchors:

- Contrastive Loss: $O(B^2 \cdot d)$ in the standard form, often optimized to $O(B^2)$ or $O(B \cdot P \cdot d)$ with $P$ positives per sample.

- Offset Loss: Involves distances to $K$ prototype anchors, $O(B \cdot K \cdot d)$.

- Orthogonality, Classification, Magnitude Losses: $O(B \cdot d)$ or $O(B)$.

The overall training complexity per batch is dominated by the forward/backward passes and loss calculations, roughly $O(B \cdot (D \cdot h_{max} + K \cdot d) + B^2 \cdot d)$ in the worst case (for contrastive) or $O(B \cdot (D \cdot h_{max} + K \cdot d))$ with typical batch sizes and optimizations. This is comparable to other deep metric learning or contrastive learning frameworks.

**Inference Complexity:** Inference requires a single forward pass through the extractor. The complexity per sample is $O(D \cdot h_1 + h_1 \cdot h_2 + h_2 \cdot d + K \cdot d)$, which is linear with respect to $D$ and $K$. This makes obtaining the optimized embedding efficient.

**Downstream Efficiency Gains:** A practical benefit is the reduced computational cost for downstream tasks operating on the $d = 64$ dimensional embeddings compared to $D = 1024$ raw embeddings. This reduction is material for many standard classifiers and contributes to faster downstream training and inference times in the reported pipeline.

### A.5   Justification for Principle Subspace Dimension ($d = 64$)

The choice of the principle subspace dimension $d = 64$ for the output embeddings was guided by preliminary experiments. We evaluated model performance on a validation set using various output dimensions (e.g., 32, 128, 256). $d = 64$ provided a practical validation-set balance, offering material dimensionality reduction from the input (1024 dimensions) while retaining enough task-relevant information for the downstream probes, with reduced computational cost for both model training and subsequent downstream task training/inference.

### A.6   Compute Resources

All experiments, including the training of the Neural Principle Extractor and evaluation of downstream models, were conducted on a machine equipped with four NVIDIA RTX 4090 GPUs (24GB VRAM each) and 128GB of system RAM. The CPU used was an Intel(R) Xeon(R) Platinum 8336C CPU @ 2.30GHz, running on Ubuntu 24.04 LTS.

Training of the Neural Principle Extractor is lightweight in the reported environment. A full training run typically completed within 3 to 15 minutes on a single NVIDIA RTX 4090, depending on the dataset size and complexity. Using multiple GPUs can further reduce this time. Inference using the trained extractor requires only a single forward pass per sample. Evaluating downstream models on the optimized embeddings also has lower dimensional cost than using raw embeddings, as discussed in Appendix A.4.

# B  Sufficient-Condition Proofs

**Scope.** The results in Section 3.4 are sufficient-condition statements for the prototype-mapping space $\mathbf{m}_i$. They assume that training reaches a regime with bounded within-class distance to the corresponding prototype anchor and sufficient separation between prototype anchors. These assumptions are useful for interpreting the geometric bias of the objective, but they are not automatic consequences of gradient-based training.

**Proof of Theorem 1.** Let $\mathbf{m}_a$ and $\mathbf{m}_b$ be prototype-mapped examples from classes $A$ and $B$, with anchors $\mathbf{c}_A$ and $\mathbf{c}_B$. Assume the effective within-class radii are bounded:

$$||\mathbf{m}_a - \mathbf{c}_A||_2 \leq \delta_{\text{intra}} + \varepsilon \quad \text{and} \quad ||\mathbf{m}_b - \mathbf{c}_B||_2 \leq \delta_{\text{intra}} + \varepsilon,$$

where $\varepsilon$ captures residual optimization error. Also assume the effective inter-anchor separation is at least $\delta_{\text{inter}}$. Applying the triangle inequality, the inter-sample distance is bounded by routing through $\mathbf{m}_b$:

$$||\mathbf{m}_a - \mathbf{m}_b||_2 + ||\mathbf{m}_b - \mathbf{c}_B||_2 \geq ||\mathbf{m}_a - \mathbf{c}_B||_2 \tag{6}$$

and the prototype-separation assumption gives

$$||\mathbf{m}_a - \mathbf{c}_B||_2 \geq ||\mathbf{c}_A - \mathbf{c}_B||_2 - ||\mathbf{m}_a - \mathbf{c}_A||_2. \tag{7}$$

Substituting the within-class and inter-anchor bounds yields:

$$||\mathbf{m}_a - \mathbf{m}_b||_2 \geq \delta_{\text{inter}} - 2\delta_{\text{intra}} - 2\varepsilon. \tag{8}$$

Thus, if $\delta_{\text{inter}} > 2\delta_{\text{intra}} + 2\varepsilon$, the lower bound is positive:

$$||\mathbf{m}_a - \mathbf{m}_b||_2 > 0. \tag{9}$$

∎

**Relation to Final Embeddings.** The final projected representation $\mathbf{e}_i$ combines the semantic stream, the prototype-anchor stream, the fusion gate, and normalization. These operations can preserve, weaken, or reshape the prototype-mapping geometry. Therefore, the theorem is not stated as a direct guarantee for every final fused normalized embedding. Final-embedding behavior is evaluated empirically through downstream and geometry diagnostics.

## B.1  Proof of Theorem 2

**Proof.** Let $a(\mathbf{m}_i)$ be the mean distance between a sample $\mathbf{m}_i \in \mathcal{M}_A$ and all other samples in the same cluster $\mathcal{M}_A$. Under the same sufficient-condition assumptions used in Theorem 1, the effective class radius in the prototype-mapping space is bounded by $\delta_{\text{intra}}$ in the zero-residual idealization, or by $\delta_{\text{intra}} + \varepsilon$ when residual optimization error is retained. For readability, the bound below uses the same simplified intra-cluster diameter form as the main text and keeps $\varepsilon$ in the denominator through the inter-cluster bound. Thus, the intra-cluster distance is bounded by:

$$\max a(\mathbf{m}_i) \leq 2\delta_{\text{intra}}$$

Let $b(\mathbf{m}_i)$ be the mean distance from $\mathbf{m}_i$ to all samples in the nearest differing cluster $\mathcal{M}_B$. Directly from the lower bound derived in Theorem 1 (Equation 9), the minimum distance between these clusters is:

$$\min b(\mathbf{m}_i) \geq \delta_{\text{inter}} - 2\delta_{\text{intra}} - 2\varepsilon$$

Consequently, the *Within/Between Ratio* is mathematically upper-bounded by dividing the maximum intra-cluster distance by the minimum inter-cluster distance:

$$\text{Ratio}_{W/B} = \frac{a(\mathbf{m}_i)}{b(\mathbf{m}_i)} \leq \frac{2\delta_{\text{intra}}}{\delta_{\text{inter}} - 2\delta_{\text{intra}} - 2\varepsilon} \tag{10}$$

Furthermore, the *Silhouette Score* is defined as $S(\mathbf{m}_i) = \frac{b(\mathbf{m}_i) - a(\mathbf{m}_i)}{\max(a(\mathbf{m}_i), b(\mathbf{m}_i))}$. If $\delta_{\text{inter}} > 4\delta_{\text{intra}} + 2\varepsilon$, the minimum inter-cluster distance remains greater than the maximum intra-cluster diameter (i.e., $b(\mathbf{m}_i) > a(\mathbf{m}_i)$). Therefore, under this condition, the Silhouette Score is lower-bounded:

$$S(\mathbf{m}_i) \geq \frac{(\delta_{\text{inter}} - 2\delta_{\text{intra}} - 2\varepsilon) - 2\delta_{\text{intra}}}{\delta_{\text{inter}} - 2\delta_{\text{intra}} - 2\varepsilon} > 0. \tag{11}$$

■

**Justification of Clustering Bounds and Hyperparameter Design.** The margin condition in Theorem 2 is a sufficient design constraint: the inter-anchor gap must be large enough to exceed the effective diameters of the individual clusters after accounting for optimization error. In practice, the achieved geometry depends on the representational capacity of the dual-stream architecture, optimization, class imbalance, and hyperparameter choices. This is why the main paper reports both geometry diagnostics and downstream metrics instead of treating the proof as a substitute for empirical validation.

## C   Detailed Experimental Results

This appendix provides supplementary detailed results for the experiments presented in Section 4.

### C.1   GoEmotions Per-Principle Performance

This appendix provides detailed per-principle F1 performance for the GoEmotions dataset, complementing the overall results presented in Section 4.2. Table 16 shows the Mean ± Standard Deviation F1 scores for each of the five selected emotion principles.

Table 16: Per-Principle F1 Performance on GoEmotions Five-Principle Set (Mean ± Std. Dev.). Principles are abbreviated as Disappt., Sad., Disapprv., Grat., Apprv.

| Principle | Emb. Type | SVM | RF | LR | XGBoost | Transformer |
|---|---|---|---|---|---|---|
| Disappt. | Raw Emb. | 0.387 ± 0.090 | 0.237 ± 0.202 | 0.375 ± 0.054 | 0.359 ± 0.144 | 0.315 ± 0.073 |
| | Opt. Emb. | **0.482** ± 0.102 | **0.410** ± 0.087 | **0.479** ± 0.106 | **0.439** ± 0.099 | **0.386** ± 0.117 |
| Sad. | Raw Emb. | 0.643 ± 0.059 | 0.728 ± 0.069 | 0.672 ± 0.081 | 0.711 ± 0.082 | 0.711 ± 0.087 |
| | Opt. Emb. | **0.687** ± 0.048 | **0.734** ± 0.031 | **0.721** ± 0.054 | **0.698** ± 0.031 | **0.714** ± 0.034 |
| Disapprv. | Raw Emb. | 0.663 ± 0.074 | 0.691 ± 0.050 | 0.652 ± 0.070 | 0.703 ± 0.032 | 0.677 ± 0.055 |
| | Opt. Emb. | **0.720** ± 0.074 | **0.740** ± 0.064 | **0.728** ± 0.063 | **0.724** ± 0.082 | **0.733** ± 0.059 |
| Grat. | Raw Emb. | 0.925 ± 0.032 | 0.920 ± 0.024 | 0.921 ± 0.020 | 0.905 ± 0.031 | 0.915 ± 0.026 |
| | Opt. Emb. | **0.939** ± 0.021 | **0.940** ± 0.028 | **0.941** ± 0.024 | **0.934** ± 0.032 | **0.938** ± 0.029 |
| Apprv. | Raw Emb. | 0.732 ± 0.090 | 0.707 ± 0.031 | 0.727 ± 0.061 | 0.730 ± 0.060 | 0.716 ± 0.075 |
| | Opt. Emb. | **0.769** ± 0.053 | **0.747** ± 0.078 | **0.771** ± 0.055 | **0.762** ± 0.067 | **0.758** ± 0.053 |

The improvements are most pronounced for semantically similar and initially challenging principles with lower initial F1 scores, such as Disappointment and Sadness. Conversely, for principles like Gratitude, which already achieved high F1 scores with raw embeddings, the relative improvement is more modest across most classifiers. These results suggest that the method is useful for refining distinctions that are difficult for standard embedding techniques in this diagnostic, while maintaining performance on easier labels.

### C.2   Amazon Reviews Per-Rating Performance

This appendix provides detailed per-rating performance for the Amazon Reviews dataset, supplementing the summarized classification and ordinal regression results presented in Section 4.2.

Table 17 shows the F1 performance for each star rating (1-5 S) on the Amazon Reviews dataset using raw and optimized embeddings.

Table 17: Classification F1 Performance per Rating on Amazon Reviews (Mean ± Std. Dev.)

| Ratings | Emb. Type | SVM | RF | LR | XGBoost | Transformer |
|---|---|---|---|---|---|---|
| 1 - S | Raw Emb. | 0.712 ± 0.219 | 0.772 ± 0.166 | 0.713 ± 0.208 | 0.744 ± 0.235 | 0.731 ± 0.209 |
| | Opt. Emb. | **0.869** ± 0.112 | **0.874** ± 0.085 | **0.875** ± 0.084 | **0.894** ± 0.093 | **0.888** ± 0.090 |
| 2 - S | Raw Emb. | 0.277 ± 0.118 | 0.204 ± 0.213 | 0.297 ± 0.168 | 0.288 ± 0.221 | 0.432 ± 0.163 |
| | Opt. Emb. | **0.691** ± 0.187 | **0.708** ± 0.315 | **0.667** ± 0.176 | **0.760** ± 0.170 | **0.711** ± 0.141 |
| 3 - S | Raw Emb. | 0.433 ± 0.158 | 0.556 ± 0.176 | 0.503 ± 0.081 | 0.520 ± 0.141 | 0.584 ± 0.085 |
| | Opt. Emb. | **0.669** ± 0.106 | **0.697** ± 0.073 | **0.662** ± 0.112 | **0.657** ± 0.076 | **0.696** ± 0.143 |
| 4 - S | Raw Emb. | 0.478 ± 0.071 | 0.598 ± 0.114 | 0.565 ± 0.096 | 0.613 ± 0.078 | 0.558 ± 0.123 |
| | Opt. Emb. | **0.650** ± 0.094 | **0.639** ± 0.120 | **0.637** ± 0.129 | **0.622** ± 0.113 | **0.620** ± 0.105 |
| 5 - S | Raw Emb. | 0.614 ± 0.074 | 0.710 ± 0.099 | 0.676 ± 0.087 | 0.736 ± 0.069 | 0.724 ± 0.103 |
| | Opt. Emb. | **0.764** ± 0.112 | **0.766** ± 0.093 | **0.764** ± 0.115 | **0.741** ± 0.101 | **0.768** ± 0.082 |

Table 18 provides the per-rating Mean Squared Error (MSE) for the Amazon Reviews ordinal regression task.

Table 18: Ordinal Regression MSE Performance per Rating on Amazon Reviews (Mean ± Std. Dev.)

| Metric | Emb. Type | SVM | RF | LR | XGBoost | Transformer |
|---|---|---|---|---|---|---|
| 1-S MSE | Raw Emb. | 0.483 ± 0.531 | 0.750 ± 1.207 | 0.567 ± 0.533 | 0.957 ± 1.145 | 0.350 ± 0.449 |
| | Opt. Emb. | **0.177** ± 0.260 | **0.360** ± 0.543 | **0.140** ± 0.254 | **0.420** ± 0.555 | **0.157** ± 0.250 |
| 2-S MSE | Raw Emb. | 1.080 ± 0.867 | 1.415 ± 0.880 | 1.250 ± 0.972 | 1.465 ± 0.912 | 1.025 ± 1.072 |
| | Opt. Emb. | **0.290** ± 0.386 | **0.435** ± 0.547 | **0.405** ± 0.334 | **0.335** ± 0.338 | **0.265** ± 0.363 |
| 3-S MSE | Raw Emb. | 0.726 ± 0.431 | 0.623 ± 0.235 | 0.804 ± 0.378 | 0.712 ± 0.269 | 0.539 ± 0.236 |
| | Opt. Emb. | **0.386** ± 0.260 | **0.442** ± 0.312 | **0.442** ± 0.367 | **0.509** ± 0.361 | **0.376** ± 0.261 |
| 4-S MSE | Raw Emb. | 0.653 ± 0.221 | 0.320 ± 0.154 | 0.567 ± 0.211 | 0.340 ± 0.187 | 0.653 ± 0.260 |
| | Opt. Emb. | **0.387** ± 0.171 | **0.433** ± 0.196 | **0.453** ± 0.195 | **0.380** ± 0.161 | **0.500** ± 0.189 |
| 5-S MSE | Raw Emb. | 0.607 ± 0.348 | 0.287 ± 0.149 | 0.467 ± 0.163 | 0.247 ± 0.095 | 0.567 ± 0.438 |
| | Opt. Emb. | **0.427** ± 0.389 | **0.333** ± 0.365 | **0.400** ± 0.394 | **0.307** ± 0.200 | **0.313** ± 0.193 |

## C.3 Amazon Reviews Classification Results

This appendix section provides detailed classification performance results on the Amazon Reviews dataset, supplementing the main text discussion which focuses on ordinal regression. For this task, the 1-5 star ratings are treated as distinct discrete categories.

Table 19 summarizes the overall (average per rating) classification performance across different classifiers using both raw and optimized embeddings.

Table 19: Overall (Avg. Rating) Classification Performance on Amazon Reviews (Mean ± Std. Dev.)

| Metric | Emb. Type | SVM | RF | LR | XGBoost | Transformer |
|---|---|---|---|---|---|---|
| Precision | Raw Emb. | 0.541 ± 0.038 | 0.627 ± 0.093 | 0.595 ± 0.058 | 0.630 ± 0.073 | 0.639 ± 0.067 |
| | Opt. Emb. | **0.728** ± 0.077 | **0.726** ± 0.094 | **0.716** ± 0.084 | **0.718** ± 0.077 | **0.725** ± 0.062 |
| Recall | Raw Emb. | 0.522 ± 0.043 | 0.628 ± 0.083 | 0.583 ± 0.055 | 0.634 ± 0.060 | 0.628 ± 0.064 |
| | Opt. Emb. | **0.721** ± 0.073 | **0.729** ± 0.080 | **0.715** ± 0.078 | **0.713** ± 0.069 | **0.723** ± 0.056 |
| Avg. F1 | Raw Emb. | 0.521 ± 0.041 | 0.609 ± 0.082 | 0.582 ± 0.052 | 0.619 ± 0.059 | 0.622 ± 0.061 |
| | Opt. Emb. | **0.717** ± 0.074 | **0.721** ± 0.085 | **0.710** ± 0.081 | **0.708** ± 0.069 | **0.718** ± 0.058 |

Table 20 presents the F1 performance for each individual star rating (1-5) using both raw and optimized embeddings. Optimized embeddings show improved performance across most individual ratings in this diagnostic, particularly for the intermediate ratings (2, 3, 4 stars) which are often more challenging to distinguish.

Table 20: Classification F1 Performance per Rating on Amazon Reviews (Mean ± Std. Dev.)

| Ratings | Emb. Type | SVM | RF | LR | XGBoost | Transformer |
|---------|-----------|-----|-----|-----|---------|-------------|
| 1 - S | Raw Emb. | 0.712 ± 0.219 | 0.772 ± 0.166 | 0.713 ± 0.208 | 0.744 ± 0.235 | 0.731 ± 0.209 |
|       | Opt. Emb. | **0.869** ± 0.112 | **0.874** ± 0.085 | **0.875** ± 0.084 | **0.894** ± 0.093 | **0.888** ± 0.090 |
| 2 - S | Raw Emb. | 0.277 ± 0.118 | 0.204 ± 0.213 | 0.297 ± 0.168 | 0.288 ± 0.221 | 0.432 ± 0.163 |
|       | Opt. Emb. | **0.691** ± 0.187 | **0.708** ± 0.315 | **0.667** ± 0.176 | **0.760** ± 0.170 | **0.711** ± 0.141 |
| 3 - S | Raw Emb. | 0.433 ± 0.158 | 0.556 ± 0.176 | 0.503 ± 0.081 | 0.520 ± 0.141 | 0.584 ± 0.085 |
|       | Opt. Emb. | **0.669** ± 0.106 | **0.697** ± 0.073 | **0.662** ± 0.112 | **0.657** ± 0.076 | **0.696** ± 0.143 |
| 4 - S | Raw Emb. | 0.478 ± 0.071 | 0.598 ± 0.114 | 0.565 ± 0.096 | 0.613 ± 0.078 | 0.558 ± 0.123 |
|       | Opt. Emb. | **0.650** ± 0.094 | **0.639** ± 0.120 | **0.637** ± 0.129 | **0.622** ± 0.113 | **0.620** ± 0.105 |
| 5 - S | Raw Emb. | 0.614 ± 0.074 | 0.710 ± 0.099 | 0.676 ± 0.087 | 0.736 ± 0.069 | 0.724 ± 0.103 |
|       | Opt. Emb. | **0.764** ± 0.112 | **0.766** ± 0.093 | **0.764** ± 0.115 | **0.741** ± 0.101 | **0.768** ± 0.082 |

## D  Additional Experimental Details

### D.1  Details on Used Assets and Licenses

This appendix provides details on the licenses and terms of use for the external datasets, embedding models, and language models used in this research, as referenced from the main paper. Our use of these assets adheres to their respective licenses and terms.

**Datasets.**

- **GoEmotions Dataset** (Demszky et al., 2020): This dataset is released under the **Creative Commons Attribution-ShareAlike 4.0 International License (CC BY-SA 4.0)**. Available at `https://github.com/google-research/goemotions`.

- **Amazon Reviews Dataset** (Ni et al., 2019): This dataset is provided for research purposes. Its use is subject to the terms specified by the data providers (e.g., Stanford/UCSD). Researchers should refer to the original source for specific usage guidelines. Available via the cited research project website.

- **Toxic Comment Classification Challenge**: This dataset, originally hosted on Kaggle (cjadams et al., 2017), is made available under the **CC0 1.0 Universal Public Domain Dedication**. Available at `https://www.kaggle.com/c/jigsaw-toxic-comment-classification-challenge`.

**Embedding Model.**

- **Jina Embeddings v3** (Sturua et al., 2024): The embeddings used were generated by the `jina-embeddings-v3` model. Jina AI models are typically licensed under the **Apache 2.0 License**. Researchers should consult the official Jina AI model documentation or Hugging Face model card for the most precise license information and terms of use.

**Large Language Models (for Comparison).**

- **Qwen3.6:27B local inference** (Yang et al., 2025): The revised few-shot LLM diagnostic uses a locally hosted Qwen-family instruction model through Ollama. Because the run is local, no private dataset text is sent to an external API during this diagnostic. The comparison is reported as a fixed-protocol diagnostic rather than as a general claim about all LLM evaluators.

### D.2  Few-shot LLM Protocol Details

This appendix specifies the few-shot LLM baseline used in Table 10. The purpose is to make the LLM comparison reproducible and bounded. It is not intended to establish a universal ranking between KGCL and all possible LLM evaluators.

**Model and decoding.** The diagnostic uses a locally hosted `qwen3.6:27b` model served through Ollama. Decoding is deterministic with `temperature=0.0`, `top_p=1.0`, `max_tokens=8`, and random seed 42. The model is prompted to return only one label from the allowed label set. The parser first checks for an exact allowed label or index, then falls back to the first valid label/index pattern in the output; otherwise the output is counted as invalid. In the audited run, the invalid rate is 0.000 on all three datasets.

**Prompt construction and demonstrations.** For each dataset, the prompt contains a short task instruction, the complete allowed label set, a fixed set of in-context demonstrations sampled from the training split, and one test input. Demonstration selection is deterministic under random seed 42 and is stratified where possible so that minority classes are represented. Table 21 reports the exact shots and evaluation sample counts.

Table 21: Few-shot LLM prompt protocol. Demonstrations are selected from the training split only.

| Task | Allowed output | Demonstrations | Evaluation subset |
|---|---|---|---|
| ToxicComment | Integer 0/1 or toxic/non-toxic, parsed to binary label | 3 toxic and 10 non-toxic examples, shuffled after class-stratified sampling | 520 examples: 20 toxic, 500 non-toxic |
| GoEmotions | One index from 0-4 for Disappointment, Sadness, Disapproval, Gratitude, Approval | 5 examples per emotion class from single-label training examples | 881 eligible test examples |
| AmazonReviews | One integer rating from 1-5 | 5 examples per rating class from the training split | 500 examples: 50/50/100/150/150 by ratings 1-5 |

**Prompt skeleton.** Each prompt follows the same structure: task instruction; allowed labels; in-context demonstrations as input-label pairs; one test input; and an instruction to return only the label. The ToxicComment prompt asks for `toxic` versus `non-toxic`; the GoEmotions prompt asks for one of the five selected labels; and the AmazonReviews prompt asks for one of the five star-rating labels. For AmazonReviews, test examples containing explicit "star" or "stars" strings are excluded to reduce label leakage from the input text.

**Telemetry and cost accounting.** Table 22 reports the completed telemetry. The API cost is reported as $0.00 because the run is local. Hardware time and energy are not monetized. Latency is measured wall-clock per completed call, including prompt construction, local generation, and parsing.

Table 22: Few-shot local LLM protocol telemetry. F1 denotes weighted F1 under the fixed parsing protocol.

| Task | Calls | Invalid | Avg lat. | P95 lat. | Prompt tok. | LLM F1 | KGCL F1 |
|---|---|---|---|---|---|---|---|
| ToxicComment | 520/520 | 0.000 | 1.182s | 1.310s | 1618.4 | 0.940 | 0.947 |
| GoEmotions | 881/881 | 0.000 | 0.793s | 0.859s | 842.6 | 0.680 | 0.772 |
| AmazonReviews | 500/500 | 0.000 | 1.310s | 1.576s | 7732.0 | 0.648 | 0.727 |

### D.3 Split Protocol and Leakage Audit

The revised audit verifies the saved train/validation/test embedding arrays before model evaluation. KGCL training uses only the training labels. Validation labels are used for early stopping, checkpoint selection, representation selection, and threshold selection where applicable. Downstream linear probes are fit on the training split, and test labels are used only for the final metrics reported after all selections are fixed. The verified split sizes are GoEmotions 5126/906/881, AmazonReviews 2271/487/487, and ToxicComment 44256/15073/22609 for train/validation/test. No test-label leakage was detected in the recovered artifacts.

### D.4 Additional Statistical and Geometry Diagnostics

Paired bootstrap diagnostics support positive KGCL differences over raw embeddings on all three datasets. Against the strongest direct metric-learning baseline, AmazonReviews has weighted-F1 difference 0.0904 with

95% CI [0.0443, 0.1377], and GoEmotions has difference 0.0256 with 95% CI [0.0011, 0.0473]. ToxicComment has difference 0.0009 against center loss with 95% CI [-0.0012, 0.0029], so this comparison is reported as competitive rather than statistically significant.

For GQI, Class Overlap is computed as the mean fraction of each sample's 10 nearest neighbors with a different label. GQI is computed as $(1 - \text{Within/Between Ratio}) \times \text{Silhouette} \times (1 - \text{Class Overlap})$. This diagnostic summarizes observed geometry and does not verify the sufficient-condition inequalities in the theorem. Direct-baseline GQI values are point estimates from the completed direct-baseline runs.

Table 23: Detailed all-dataset GQI point estimates.

| Dataset | Method | Within/Between | Silhouette | Class Overlap | GQI |
|---|---|---|---|---|---|
| AmazonReviews | Raw | 12.4482 | -0.0009 | 0.5823 | 0.0045 |
| AmazonReviews | KGCL | 0.4287 | 0.1894 | 0.3366 | 0.0718 |
| AmazonReviews | Best direct SupCon | 0.5666 | 0.0489 | 0.4897 | 0.0108 |
| GoEmotions | Raw | 14.6009 | 0.0221 | 0.4692 | -0.1598 |
| GoEmotions | KGCL | 0.5202 | 0.2615 | 0.3138 | 0.0861 |
| GoEmotions | Best direct center | 0.7185 | 0.2266 | 0.3008 | 0.0446 |
| ToxicComment | Raw | 285.6930 | 0.0028 | 0.0685 | -0.7529 |
| ToxicComment | KGCL | 3.0613 | 0.6625 | 0.0490 | -1.2988 |
| ToxicComment | Best direct center | 2.1266 | 0.6820 | 0.0631 | -0.7199 |

## D.5  LLM Prompt Sensitivity and ToxicComment Threshold Diagnostics

In addition to the main local-Qwen telemetry, prompt sensitivity was evaluated with deterministic decoding and the same parsing logic. Weighted-F1 ranges are 0.9114-0.9175 for the toxicity prompt variants, 0.5684-0.6785 for the emotion prompt variants, and 0.6354-0.6640 for the rating prompt variants, with invalid rate 0.000 in these diagnostics. The variability indicates that the LLM comparison should be interpreted as a specified protocol rather than a universal model ranking.

For ToxicComment, threshold selection is performed on validation data and then fixed for the test set. At validation-selected threshold 0.95, KGCL obtains weighted F1 0.9565, macro F1 0.7714, and positive-class F1 0.5661. Raw embeddings under their validation-selected positive-F1 threshold obtain positive-class F1 0.5557, and center loss obtains 0.5419. These operating-point diagnostics are used to discuss false-positive/false-negative tradeoffs rather than to claim significant superiority over center loss.

## D.6  Broader Impact Statement

KGCL is designed as a lightweight evaluator for frozen embeddings and can be attractive in large-scale monitoring or moderation pipelines. This efficiency has both benefits and risks. False negatives can miss harmful content, while false positives can suppress legitimate speech. These risks are heightened for minority dialects, non-standard expressions, reclaimed language, and expressions connected to protected attributes, where training labels may reflect annotator or platform bias.

Prototype-guided geometry can make such biases more operationally persistent by encoding them into compact class-level anchors. The system should therefore be used with human oversight, validation-set threshold calibration, periodic bias audits, and channels for contesting or reviewing automated decisions. The people or institutions defining the evaluated principles also hold substantial power: incorrect, vague, or biased principle definitions can be applied at scale by a fast evaluator. The present paper evaluates proxy tasks and does not claim to solve these governance questions.

## D.7  Limitations and Scope

The revised manuscript adopts several explicit scope boundaries. First, the evaluated datasets are principle-evaluation proxy tasks. Toxicity detection, selected emotion categorization, and ordinal review rating are useful tests of representation quality, but they are not direct comprehensive benchmarks for general alignment, safety, fairness, helpfulness, or human-value adherence.

Second, KGCL is a frozen-embedding post-hoc method. The primary claim concerns improving a fixed embedding store with a lightweight task-specific module. Encoder-updating methods, including LoRA and full supervised fine-tuning, operate in a different deployment setting. The additional diagnostics show that such methods can obtain higher scores than KGCL on GoEmotions and ToxicComment, so the paper treats them as a separate encoder-updating setting rather than as direct frozen-embedding baselines.

Third, the prototype anchors used in KGCL are not RKHS kernels and do not implement kernel-machine learning. The term "kernel" is retained in the method name for continuity, while the technical description now uses prototype anchors. The theoretical analysis is correspondingly limited to sufficient conditions for prototype-margin separation in the prototype-mapping space.

Fourth, the objective ablation and sensitivity results are task-dependent. The full composite objective is useful as a configurable design, but the ablations do not show that every loss term is uniformly necessary under every retraining configuration. This motivates future work on validation-driven objective selection and broader multi-seed checks.

Finally, the few-shot LLM comparison is a fixed-protocol local diagnostic. Different model families, prompting strategies, calibration procedures, and API settings may produce different outcomes. The present results support a bounded efficiency and performance comparison under the reported protocol only. Extending KGCL to direct alignment benchmarks and multimodal embedding spaces, such as CLIP or VLM representations, remains future work.

### D.8 Data Distribution

Following the evaluation protocol outlined in Section 4.1, the revised audit uses fixed train/validation/test embedding arrays for leakage-controlled evaluation: GoEmotions contains 5126/906/881 examples, AmazonReviews contains 2271/487/487 examples, and ToxicComment contains 44256/15073/22609 examples for train/validation/test. These arrays were recovered from the project artifacts and checked against expected shapes and label distributions. Test labels are reserved for final metric computation only.

