# OpenReview forum: "Structuring Semantic Embeddings for Principle Evaluation: A Kernel-Guided Contrastive Learning Approach"
_TMLR — Under review for TMLR_

### Review · Reviewer_Ke7u · 2026-05-22

**Summary Of Contributions:**

This paper studies post-hoc principle evaluation, where text is evaluated according to predefined principles such as safety, toxicity, emotion, sentiment, or rating intensity. The authors argue that general-purpose text embeddings often entangle semantically similar but principle-distinct examples, making downstream classification difficult. To address this issue, the paper proposes Kernel-Guided Contrastive Learning, or KGCL, a lightweight module built on top of frozen text embeddings. KGCL projects high-dimensional embeddings into a lower-dimensional task-specific subspace using learnable “kernel” anchors, attention-based mapping, and a composite objective consisting of supervised contrastive loss, offset loss, orthogonality loss, and magnitude loss. The paper also provides theoretical bounds on class separation and clustering quality, and evaluates the method on GoEmotions, Amazon Reviews, and Toxic Comment Classification datasets.

The reported results suggest that the learned representations improve downstream classification and regression performance over raw embeddings and some contrastive baselines. The paper further claims that KGCL can outperform few-shot LLM-based evaluators while being substantially more efficient.

**Additional Comments:**

See above.

**Audience:**

Yes

**Audience Explanation:**

The general topic is relevant to TMLR’s audience. Efficient post-hoc evaluation, task-specific representation learning, limitations of general-purpose embeddings, and lightweight alternatives to LLM-as-a-judge systems are all timely topics.

**Broader Impact Concerns:**

N/A.

**Claims And Evidence:**

No

**Claims Explanation:**

The paper addresses an interesting and practically relevant problem, and the empirical results indicate that the proposed projection module can improve performance over raw embeddings on several datasets. However, I do not think the current evidence is sufficient to support the strength of the paper’s claims.

First, the core novelty is not clearly established. Although the method is described as “kernel-guided,” the learnable kernels appear to function essentially as class prototypes, anchors, or centroids. The paper does not provide a formal connection to classical kernel methods, kernel machines, or RKHS-style kernel learning. Therefore, the terminology seems potentially misleading, and the method appears closer to prototype-guided supervised contrastive learning with margin regularization.

Second, the overall objective appears over-engineered. The final loss combines supervised contrastive loss, offset loss, orthogonality loss, and magnitude loss. While each term is individually motivated, the ablation study does not convincingly demonstrate that all components are necessary. Several reduced variants achieve performance close to the full model, especially on GoEmotions. This weakens the claim that the complete composite objective is essential. The paper would need stronger ablations, interaction analysis among loss terms, and hyperparameter sensitivity studies to justify the complexity of the proposed objective.

Third, the theoretical contribution is limited. The stated bounds mainly formalize the intuitive result that if samples are constrained to be close to their own prototype and far from other prototypes, then clusters become separated. This is mathematically valid, but it relies on strong assumptions such as sufficient parameterization and near-zero optimization error. It is also not clear how much the theory distinguishes KGCL from existing metric learning, center loss, prototypical learning, or margin-based representation learning methods. In addition, the transition from bounds on the prototype mapping space to guarantees on the final normalized fused embeddings is not fully convincing, since the semantic stream, gating, and normalization may alter the geometry.

Fourth, the experimental comparisons are incomplete. The paper compares against raw embeddings, SimCSE-style contrastive learning, and few-shot LLM evaluators, but it lacks comparisons with more direct and relevant metric-learning baselines, such as center loss, triplet loss, supervised contrastive learning with class prototypes, prototypical networks, ArcFace/CosFace-style margin losses, or simple prototype classifiers. These baselines are important because the proposed method is very close in spirit to prototype- and margin-based metric learning.

Fifth, the comparison with LLM-based evaluators is not sufficiently convincing. The paper claims that KGCL outperforms few-shot LLMs, but the LLM evaluation protocol is not described in enough detail. Prompt design, number and selection of demonstrations, calibration, decoding settings, and whether prompts were tuned on validation data can all substantially affect performance. Without a more rigorous protocol, the claim that KGCL outperforms LLM-as-a-judge approaches seems too strong.

Finally, the framing around “principle evaluation” is broader than what the experiments demonstrate. The datasets used are standard emotion, review rating, and toxicity classification datasets. These are useful proxies, but they do not fully validate the broader claims about evaluating human values such as safety, fairness, helpfulness, or alignment of generated text. The paper should either provide experiments on more direct principle-evaluation settings or narrow its claims.

**Requested Changes:**

1. Clarify the terminology and novelty of “kernel-guided” learning. If the learnable kernels are essentially prototypes or class anchors, the paper should either use that terminology or provide a formal justification for the kernel terminology.

2. Add comparisons with stronger and more directly relevant metric-learning baselines, including center loss, triplet loss, prototypical networks, supervised contrastive learning with prototypes, prototype classifiers, and margin-based losses such as ArcFace or CosFace.

3. Simplify or better justify the composite objective. The current ablation study does not convincingly show that all loss terms are necessary. The authors should provide stronger ablations, loss-term interaction analysis, and sensitivity studies for the loss weights and margin hyperparameters.

4. Strengthen the theoretical analysis. The current theory largely follows from the imposed prototype-margin constraints. The authors should clarify what is theoretically new beyond standard margin-based metric learning, and should more carefully justify whether the bounds apply to the final fused and normalized embeddings.

5. Provide a more rigorous LLM baseline protocol. The paper should report prompts, number of demonstrations, demonstration selection strategy, decoding settings, calibration method, and whether prompts were tuned. Otherwise, the comparison with LLM evaluators is difficult to interpret.

6. Narrow or better support the “principle evaluation” framing. The current datasets are useful but mostly standard classification or regression benchmarks. The authors should either include more direct post-hoc alignment/principle-evaluation tasks or reduce the scope of their claims.

7. Improve the presentation. Some parts of the paper use inflated terminology such as “decision boundary sculpting,” “geometric shield,” and “topological isolation,” which makes the contribution appear stronger than what is technically demonstrated. Figure 1 is also more illustrative than explanatory and should be replaced with a cleaner diagram that maps directly to the architecture and loss components.

---

> ### Author Response · Authors · 2026-06-23
> **Response to Reviewer Ke7u (1)**
>
> We sincerely thank the reviewer for the detailed suggestions on terminology, baseline choice, objective complexity, theory, LLM protocol, task framing, and presentation. These comments helped us revise the manuscript toward a narrower and more reproducible claim. We respond to each requested change below and have updated the paper accordingly (changes are highlighted in red in the revised manuscript).
>
> ### Response to Requested Change 1: Clarify the terminology and novelty of "kernel-guided" learning
>
> We thank the reviewer for raising this terminology issue.
>
> The objects called kernels in KGCL function technically as learnable class-level prototype anchors. The contribution is prototype-guided regularization of frozen embeddings, not RKHS kernel learning, kernel machines, or a kernel trick.
>
> We have revised the manuscript to define these objects as learnable prototype anchors at first use. The revised manuscript retains "Kernel-Guided Contrastive Learning" as the method name for continuity, while the technical description consistently uses "prototype anchors" and "prototype-guided regularization." These revisions are shown in the red-highlighted text in the Abstract, Introduction, Section 3.1, and Appendix A.2.
>
> ### Response to Requested Change 2: Add comparisons with stronger and more directly relevant metric-learning baselines
>
> We thank the reviewer for emphasizing the need for direct metric-learning comparisons.
>
> Because KGCL operates on frozen embeddings, the fairest primary comparison is against methods trained under the same frozen-embedding and linear-probe protocol. The revised Experiments section therefore includes SupCon projection head, center loss, triplet loss, prototype classifier, ArcFace, and CosFace.
>
> As shown in Table 6, KGCL obtains 0.727/0.772/0.953 weighted F1 on AmazonReviews/GoEmotions/ToxicComment, while the strongest direct baselines obtain 0.628/0.764/0.952. The revised text reports these results as a strong frozen-embedding comparison and explicitly treats smaller margins as dataset-dependent.
>
> ### Response to Requested Change 3: Simplify or better justify the composite objective
>
> We thank the reviewer for the suggestion to better justify the composite objective.
>
> The objective is intended as a configurable prototype-guided regularization design: supervised contrastive loss establishes class-level clustering, the offset loss supplies explicit prototype margins, orthogonality reduces redundancy between streams, and the optional magnitude term handles ordinal structure. The revised manuscript does not claim that every component is uniformly necessary in every task.
>
> We have added same-provenance objective ablations, loss-weight sensitivity, and interaction diagnostics in the Experiments section. These results are reported in the red-highlighted Tables 13 and 14 and show task-dependent component effects. They motivate the revised wording: KGCL is a structured and configurable objective, not a proof that a fixed loss combination is always optimal.
>
> ### Response to Requested Change 4: Strengthen the theoretical analysis
>
> We thank the reviewer for asking us to clarify the connection between the theorem and the final representation.
>
> The intended theoretical contribution is a sufficient-condition analysis in the prototype-mapping space. The final representation `e = normalize(alpha s + (1-alpha)m)` includes semantic fusion, gating, and normalization, which may preserve or reshape the margins encouraged in the prototype-mapping stream.
>
> We have revised the Method section and Appendix proofs accordingly. The revised theorem statements apply to the prototype-mapping space with residual error terms, and the final fused embeddings are evaluated empirically through task performance and geometry diagnostics rather than claimed to be covered by an unconditional theoretical guarantee. These revisions are shown in the red-highlighted text in Section 3.4 and Appendix B.

---

> ### Author Response · Authors · 2026-06-23
> **Response to Reviewer Ke7u (2)**
>
> ### Response to Requested Change 5: Provide a more rigorous LLM baseline protocol
>
> We thank the reviewer for requesting a more rigorous LLM baseline protocol.
>
> The revised comparison is intentionally protocol-specific. KGCL is a lightweight frozen-embedding evaluator, while LLM prompting depends on model choice, demonstrations, decoding, parsing, and cost assumptions.
>
> We have added an auditable local `qwen3.6:27b` few-shot protocol in the Experiments section and Appendix D.2, `Few-shot LLM Protocol Details`. The appendix reports prompt skeletons, shot counts, demonstration selection from the training split, random seed, deterministic decoding, parsing rules, invalid-output handling, calls, latency, token counts, and cost status. The revised text avoids a broad superiority claim over all LLM evaluators. These additions are shown in the red-highlighted Table 10 and Appendix D.2.
>
> ### Response to Requested Change 6: Narrow or better support the "principle evaluation" framing
>
> We thank the reviewer for this important framing suggestion.
>
> Our intended setting is post-hoc evaluation using proxy tasks where labels reflect toxicity, selected emotions, or ordinal rating intensity. These tasks are useful for evaluating whether frozen embeddings can be restructured around task-relevant distinctions, but they are not direct comprehensive benchmarks for general alignment, fairness, helpfulness, or human-value adherence.
>
> We have revised the manuscript to use the phrase "principle-evaluation proxy tasks" and to list direct alignment, safety, fairness, and generated-text evaluation benchmarks as future work. These revisions are shown in the red-highlighted scope statements in the Abstract, Introduction, Experimental Setup, Conclusion, and Appendix D.7, Limitations and Scope.
>
> ### Response to Requested Change 7: Improve the presentation
>
> We thank the reviewer for the presentation suggestions.
>
> The revised manuscript uses more technical and less rhetorical terminology. The contribution is described as prototype-guided projection and margin-regularized representation learning for frozen embeddings, rather than as a broad geometric or topological guarantee.
>
> Specifically, phrases that could imply a formal final-embedding separation guarantee were revised to "downstream linear-probe performance," "empirical downstream-probe behavior," or "geometry diagnostics" where the claim is empirical. The Method section reserves theorem language for sufficient conditions in the prototype-mapping stream, and the Experiments section evaluates final embeddings through task metrics, bootstrap intervals, GQI, and qualitative visualization.
>
> We have revised Figure 1 into an architecture and loss diagram. It now shows the frozen encoder, semantic MLP stream, prototype-anchor attention stream, fusion and normalization, and the supervised contrastive, offset, orthogonality, and magnitude losses. We also revised wording throughout the manuscript to align with this more precise presentation. These revisions are shown in the red-highlighted Figure 1 caption and surrounding Introduction text.

---

### Review · Reviewer_pZ63 · 2026-05-26

**Summary Of Contributions:**

This paper proposes KGCL, a lightweight framework that projects frozen general-purpose text embeddings into a task-specific, principle-aligned subspace. The key idea is to use learnable prototype kernels and S contrastive losses to reduce representational entanglement between semantically similar but principle-distinct texts. Experiments on emotion, review-rating, and toxicity datasets show improved downstream performance over raw embeddings and some contrastive/LLM-based baselines.

The main strengths are its practical motivation, modular design, and efficient use of frozen encoders. However, the method seems closely related to existing metric learning, supervised contrastive learning, and prototype-based losses, so the novelty is not fully clear. Stronger baselines and more direct principle-alignment benchmarks would make the claims more convincing.

**Audience:**

Yes

**Audience Explanation:**

At least some readers in the TMLR audience would likely be interested in the paper because it addresses a relevant problem: improving the reliability and efficiency of post-hoc evaluation using frozen text embeddings. The idea of explicitly reshaping embedding geometry for principle-specific evaluation may be useful to researchers working on representation learning, alignment evaluation, text classification, and efficient alternatives to LLM-as-a-judge systems.

However, the level of interest may be limited unless the paper better clarifies its novelty over existing metric learning, supervised contrastive learning, and prototype-based representation methods. The findings are practically relevant, but the current framing may overstate the theoretical and conceptual contribution.

**Claims And Evidence:**

No

**Claims Explanation:**

The claims are only partially supported. The empirical results suggest that KGCL can improve downstream performance over raw embeddings and some contrastive baselines, but the evidence is not fully convincing for the broader claims made in the paper. In particular, the proposed method appears closely related to existing metric learning, supervised contrastive learning, center/prototype losses, and margin-based representation learning, but the paper does not sufficiently clarify its novelty over these prior approaches.

I am also not convinced by the theoretical contribution. The two theorems mainly formalize that if intra-class distances are bounded and inter-class prototype distances are enforced to be sufficiently large, then the resulting clusters will be separated and have favorable clustering metrics. These results seem to follow directly from the imposed margin constraints and triangle-inequality-style reasoning, rather than providing a fundamentally new theoretical insight. Moreover, the guarantees rely on strong assumptions such as sufficient parameterization and successful optimization, which may not hold in practice.

The experimental evidence would be stronger with more direct comparisons to standard metric-learning and prototype-based baselines, such as center loss, triplet/margin loss, supervised contrastive learning with a projection head, prototypical classifiers, PEFT methods, and supervised embedding fine-tuning. The LLM comparison is also not fully convincing without more detailed prompting, calibration, and inference-cost analysis. Therefore, while the method is practically interesting, the current evidence does not fully support the strength and generality of the paper’s claims.

**Requested Changes:**

1. Clarify the novelty over existing metric-learning methods.: The paper should clearly distinguish KGCL from supervised contrastive learning, center loss, triplet/margin loss, prototypical networks, and prototype-based classifiers. The current method appears closely related to these established approaches.

2. Moderate or better justify the theoretical claims.: The two theorems seem to follow directly from the imposed intra-class and inter-class margin constraints. The paper should explain what is theoretically new, and clarify the assumptions under which the guarantees hold.

3. Add stronger baselines.: The experiments should include supervised contrastive learning with a projection head, center loss, triplet/margin loss, prototypical classifiers, PEFT/LoRA adaptation, and supervised embedding fine-tuning.

4. Clarify the LLM comparison setting.: The paper should provide prompt templates, number of shots, decoding settings, calibration/parsing rules, and inference cost/latency. Otherwise, the claim that KGCL outperforms LLM-based evaluators is not fully convincing.

5. Discuss applicability beyond text embeddings, especially to multimodal or image embeddings.: Would strengthen the work. Since KGCL operates on frozen embeddings and learns a task-specific geometric subspace, it may be applicable to image or multimodal representations, such as CLIP/VLM embeddings. The authors should discuss whether the same prototype-guided geometric structuring can be used for image-based safety evaluation, visual content moderation, or multimodal principle evaluation.

---

> ### Author Response · Authors · 2026-06-23
> **Response to Reviewer pZ63 (1)**
>
> We sincerely thank the reviewer for the thoughtful comments on novelty, theoretical scope, baseline strength, LLM comparison, and applicability beyond text. These suggestions helped us make the manuscript more precise and better aligned with the evidence. We respond to each requested change below and have updated the paper accordingly (changes are highlighted in red in the revised manuscript).
>
> ### Response to Requested Change 1: Clarify the novelty over existing metric-learning methods
>
> We thank the reviewer for pointing out that KGCL should be positioned relative to supervised contrastive learning, center loss, triplet or margin losses, prototypical networks, and prototype classifiers.
>
> Our intended contribution is not to present KGCL as unrelated to metric learning. Rather, KGCL studies a frozen-embedding post-hoc setting in which prototype-anchor attention, supervised contrastive separation, offset-based prototype margins, and stream regularization are combined to reshape a fixed text-embedding store for principle-evaluation proxy tasks.
>
> We have revised the manuscript to make this positioning explicit. The revised manuscript now describes standard metric-learning objectives as the closest methodological family and presents KGCL as a prototype-guided regularization framework rather than as a separate or incompatible paradigm. These revisions are shown in the red-highlighted text in the Introduction, Related Work, and Section 3.1 problem formulation.
>
> ### Response to Requested Change 2: Moderate or better justify the theoretical claims
>
> We thank the reviewer for the suggestion to make the theoretical claim more carefully scoped.
>
> The intended role of the theory is to formalize the geometric bias encouraged by the prototype-margin objective: bounded within-class spread, sufficient inter-prototype separation, small optimization residual, and sufficient capacity in the prototype-mapping stream. The theory is not intended as an unconditional guarantee for every final fused normalized embedding produced by training.
>
> Following this scope clarification, we also revised wording that could be read as a formal final-embedding separability statement. The manuscript now uses downstream linear-probe performance and empirical geometry diagnostics for the final embedding, while reserving theorem language for sufficient conditions in the prototype-mapping stream.
>
> We have revised Theorem 1 and Theorem 2 in the Method section and the Appendix proofs as sufficient-condition statements for the prototype-mapping space. The revised discussion explicitly states that the final fused normalized embedding is evaluated empirically through downstream metrics, bootstrap intervals, and geometry diagnostics. These revisions are shown in the red-highlighted text in Section 3.4 and Appendix B, Sufficient-Condition Proofs.
>
> ### Response to Requested Change 3: Add stronger baselines
>
> We thank the reviewer for this important suggestion. Direct supervised metric-learning and prototype baselines are the most relevant comparison for the frozen-embedding setting studied in the paper.
>
> The revised manuscript evaluates KGCL against supervised contrastive projection head, center loss, triplet loss, prototype classifier, ArcFace, and CosFace under the same frozen `jina-embeddings-v3` and linear-probe protocol. The main results are reported in the red-highlighted Table 6, with paired bootstrap intervals in the red-highlighted Table 7.
>
> | Dataset | RAW | KGCL | Best direct baseline | Method |
> | --- | --- | --- | --- | --- |
> | AmazonReviews | 0.589 | **0.727** | 0.628 | SupCon projection head |
> | GoEmotions | 0.726 | **0.772** | 0.764 | Prototype classifier |
> | ToxicComment | 0.915 | **0.953** | 0.952 | Center loss |
>
> These results show that KGCL is strong in the frozen-embedding protocol, with the clearest margin on AmazonReviews and smaller margins on GoEmotions and ToxicComment. We also added a separate supervised adaptation diagnostic table for a frozen residual adapter, encoder-LoRA, and full encoder fine-tuning. These results are presented separately because encoder-updating methods address a different deployment setting from KGCL's frozen-embedding post-hoc setting.

---

> ### Author Response · Authors · 2026-06-23
> **Response to Reviewer pZ63 (2)**
>
> ### Response to Requested Change 4: Clarify the LLM comparison setting
>
> We thank the reviewer for pointing out that the LLM comparison should be reproducible and protocol-specific.
>
> The intended comparison is between a lightweight frozen-embedding evaluator and a fixed few-shot local-model diagnostic, not a universal claim about all possible LLM evaluators. For this reason, the revised manuscript reports the LLM baseline as an audited protocol with model, prompt construction, demonstrations, decoding, parsing, invalid-output handling, sample counts, latency, token counts, and cost status.
>
> Specifically, the Experiments section reports a local `qwen3.6:27b` few-shot run through Ollama with deterministic decoding. The run completed 520/520 ToxicComment calls, 881/881 GoEmotions calls, and 500/500 AmazonReviews calls with zero invalid predictions. Under this protocol, the LLM obtains weighted F1 of 0.940, 0.680, and 0.648, compared with KGCL's 0.947, 0.772, and 0.727. The red-highlighted Table 10 reports the telemetry, Appendix D.2 reports the prompt protocol, and Appendix D.5 reports prompt sensitivity.
>
> ### Response to Requested Change 5: Discuss applicability beyond text embeddings
>
> We thank the reviewer for this helpful suggestion. Multimodal embeddings are a natural future direction for a frozen-embedding geometric method.
>
> The current manuscript validates KGCL on text embeddings only. The broader idea is that prototype-guided regularization may be applicable when frozen CLIP or VLM embeddings are used with labeled safety, moderation, or principle-evaluation proxy tasks, but such an extension requires modality-specific prototype design and direct multimodal benchmarks.
>
> We have revised the Discussion and Limitations to state this boundary clearly. The manuscript now presents multimodal and image-embedding applications as future work rather than as an experimentally established claim. These scope revisions are shown in the red-highlighted future-work discussion and Appendix D.7, Limitations and Scope.

---

### Review · Reviewer_h6mG · 2026-06-15

**Summary Of Contributions:**

This paper proposes Kernel-Guided Contrastive Learning, or KGCL. KGCL is a lightweight module. It takes frozen general-purpose embeddings as input and projects them into a low-dimensional task-specific subspace. The model uses a dual-stream architecture made of a semantic basis stream and a prototype mapping stream. It uses learnable prototype kernels as geometric anchors for each principle. The paper also introduces an objective function that combines supervised contrastive loss, offset loss, orthogonality loss, and magnitude loss for ordinal tasks. The paper presents its contributions as three main points: a modular architecture that does not update the frozen large-scale encoder, a geometry-aware training objective that separates semantically similar but principle-distinct texts, and empirical improvements across diverse datasets.

Strengths:

The problem is clearly stated. The paper focuses on the unclear fine-grained decision boundaries that appear when general-purpose embeddings are used for post-hoc evaluation. It also evaluates several task types. In addition to comparisons with raw embeddings, SimCSE, and few-shot LLMs, the paper uses GoEmotions, Amazon Reviews, and the Toxic Comment Classification Challenge. In the experiments, the authors project 1024-dimensional jina-embeddings-v3 embeddings into a 64-dimensional subspace. They evaluate the method on a five-emotion subset of GoEmotions, 1-5 star ratings in Amazon Reviews, and an imbalanced Toxic Comment setting.

Weaknesses:

First, the theoretical guarantees are mainly conditional geometric results. In effect, they say that if the loss is zero, if there is enough parameterization, and if the specified margin conditions hold, then the related distance bounds hold. They do not sufficiently show that the claimed linear separability of the final extracted embeddings or the claimed strict bounds are actually guaranteed. Second, the experimental setup is not reproducible enough. It is unclear whether KGCL was trained only on the training split of each fold, how the hyperparameters were chosen, and what the exact LLM prompting setup was. Third, the ablation study is limited to parts of the offset and contrastive losses. It does not clearly separate the individual effects of the dual-stream design, attention, prototype kernels, orthogonality loss, and magnitude loss. Fourth, the evidence is not enough for strong statements such as consistently outperforms LLMs, empirically validates the geometric guarantees, and statistically significant.

**Additional Comments:**

The central problem setting of the paper is reasonable, and the direction of reconstructing frozen embeddings into principle-aware representations is useful. Some experimental results, especially Amazon Reviews classification/regression and per-principle improvements on GoEmotions, suggest that the proposed method may work well as a labeled projection method. In the per-principle and per-rating results in Appendix C, it is interesting that the improvements are large for difficult categories such as Disappointment in GoEmotions and 2-star or 3-star ratings in Amazon Reviews.

However, in its current form, the paper seems to wrap a good empirical idea in overly strong theoretical guarantees and overly broad generalization claims. The theory should be limited to a geometric consequence that holds when learning ideally satisfies the margin constraints. Guarantees for the final embedding and the downstream classifier need to be shown separately. The experiments need major strengthening in reproducibility, fair comparison, statistical testing, and ablation.

**Audience:**

Yes

**Audience Explanation:**

TMLR readers would likely be interested in this paper. The reason is that representation entanglement in frozen large embedding models used for post-hoc evaluation is directly related to practical problems such as LLM evaluation, safety evaluation, content moderation, and large-scale monitoring with lightweight classifiers.

**Broader Impact Concerns:**

This paper is related to content moderation and safety evaluation, so misclassification can have social effects. Especially in an imbalanced and sensitive setting such as Toxic Comment, there are two kinds of risks: missing the minority class, and over-detecting non-toxic speech as toxic, which may suppress expression. At the end, the paper says that possible misuse of alignment technology and amplification of inherent biases need careful consideration, but it does not provide a concrete analysis.

The paper should add or expand a Broader Impact Statement. At minimum, it should discuss the effects of false positives and false negatives, the possibility that training-data bias is fixed into the prototype geometry, the effects on minority dialects, non-standard expressions, and expressions related to protected attributes, the need for human oversight in moderation deployment, and the power held by the people who define the principles being evaluated. Because KGCL aims to be a lightweight and high-throughput evaluator, there is also a risk that incorrect principle definitions or biased labels could be applied at large scale.

**Claims And Evidence:**

No

**Claims Explanation:**

Some narrow empirical claims are supported by the numbers in the tables. For example, on GoEmotions, the F1 score of Logistic Regression improves from $0.726 \pm 0.031$ with raw embeddings to $0.776 \pm 0.032$ with optimized embeddings. Similar improvements are reported for many other classifiers. For ordinal regression on Amazon Reviews, optimized embeddings are better than raw embeddings for many results in MSE, RMSE, and $R^2$. On Toxic Comment, minority F1 also improves for some classifiers.

However, the main claims are not supported well enough. The most important problem is the theory. Theorem 1 and Theorem 2 derive distance bounds and a Silhouette lower bound in the $m_i$ space under conditions such as sufficient parameterization, zero residual, and $\delta_{\mathrm{inter}} > \delta_{\mathrm{intra}}$ or $\delta_{\mathrm{inter}} > 3\delta_{\mathrm{intra}}$. But the final representation used for downstream classification is $e_i = \tilde{e}_i / \|\tilde{e}_i\|_2$, where $\tilde{e}_i = \alpha s_i + (1-\alpha)m_i$. Also, the offset constraint in the loss is mainly imposed on the distance between $m_i$ and the prototypes. The paper says that $\alpha$ gives priority to $m_i$, and that $L_2$ normalization is topology-preserving, so the margin in $m_i$ determines the linear separability of $e_i$. This is not enough as a proof. In general, $L_2$ normalization does not preserve Euclidean margins. The paper also does not show conditions under which the term $\alpha s_i$ does not break the bound.

The experimental validation of Theorem 2 is also not enough. The paper reports the Within/Between Ratio, Silhouette Score, Class Overlap, and GQI on Amazon Reviews, and says that KGCL is better than raw embeddings and SimCSE. In Table 6, KGCL has a Within/Between Ratio of $0.358$, a Silhouette Score of $0.203$, and a GQI of $0.0975$, which are better than the standard supervised baseline. However, this does not verify that the theoretical inequalities are actually satisfied for the defined $\delta_{\mathrm{intra}}$ and $\delta_{\mathrm{inter}}$. Also, GQI is a composite metric introduced in this paper. The computation method for Class Overlap and the confidence intervals are not clear. For SimCSE's GQI, if we plug the Table 6 values $1.010$, $0.010$, and $0.491$ into $(1-\mathrm{Within/Between}) \times \mathrm{Silhouette} \times (1-\mathrm{Overlap})$, the result does not seem to match the reported value $-0.0005$. The paper should explain the numerical processing or rounding.

The experimental evidence is also not clear enough for the claims. The paper reports all metrics as mean $\pm$ standard deviation over 10-fold cross-validation. However, it is not clearly stated whether KGCL itself was trained only on the training split of each fold, or whether optimized embeddings were first created using the whole dataset and then downstream classifiers were evaluated by cross-validation. If the latter is true, label information used in representation learning leaks into the test fold, and the results are overestimated. Appendix D.2 says that the authors used 6,857 samples for GoEmotions, 3,028 samples for Amazon Reviews, and 81,948 samples for the resampled Toxic Comment dataset, and that they dynamically split the data according to a 10-fold protocol. Still, the nesting relation between representation learning and downstream evaluation is not clear.

The claims about statistical significance are not supported by the evidence shown. For Toxic Comment, the main text says that optimized embeddings give statistically significant improvements in both Average F1 and Minority F1 across all classifiers. But in Table 3, the Avg. F1 of Random Forest is about $0.949$ for both raw and optimized embeddings, and the standard deviations overlap. The Avg. F1 of the Transformer also improves only slightly, from $0.956$ to $0.959$. The paper does not show p-values, paired tests, confidence intervals, or tests on fold-wise differences.

The comparison with LLMs is also insufficient. Table 5 reports that KGCL + Transformer outperforms grattafiori2024llama, DeepSeek-chat-v3, and Gemini-2.5-pro on GoEmotions F1, Amazon MSE, and Toxic Avg. F1. However, the paper does not describe the full few-shot prompts, the number of shots, how demonstration examples were selected, temperature, output parsing, the number of evaluation samples, failure handling, or variance. Therefore, the broad claim that the method consistently outperforms massive generative LLMs is not convincing based only on the current description.

Finally, the ablation study does not support the claims well enough. In Table 7, the average F1 is $0.78$ for full KGCL, $0.77$ for Without Offset Loss, $0.77$ for Only Offset Loss, and $0.77$ for Without Contrastive Loss. This does not clearly show that offset loss is mandatory. For Disappointment, the score drops from $0.49$ to $0.44$, but the difference in average performance is small. Because there is no ablation for orthogonality loss, magnitude loss, prototype initialization, attention mechanism, semantic basis stream, or the $\alpha$ gate, it is unclear which proposed components are truly necessary.

**Requested Changes:**

The following changes are critical for the acceptance decision.

1. The paper needs to make the theoretical guarantee more rigorous. The current Theorem 1 and Theorem 2 give conditional bounds in the $m_i$ space. They do not guarantee the margin, linear separability, or Silhouette bound of the final embedding $e_i$. Since the paper uses $e_i = \mathrm{normalize}(\alpha s_i + (1-\alpha)m_i)$ as the actual output, it needs to state sufficient conditions under which $\alpha s_i$ and normalization preserve the margin. The theorems should then be reformulated under those conditions.

2. The paper needs to resolve the inconsistency in the formulation of ordinal modeling. In the main text, the paper says that ordinal regression enforces a monotonic progression along kernel magnitudes. At the same time, prototype kernels are defined on $S^{d-1}$, and Appendix A.2 says that ordinal prototypes are initialized as an angular progression on the unit hypersphere. Also, the magnitude loss matches $\|m_i\|_2$ to $I(y_i)\|c_{y_i}\|_2$. The paper should mathematically clarify the relation among unit-norm prototypes, angular progression, and magnitude-based ordinal loss. It should also clearly state in which space ordinal intensity is represented.

3. The cross-validation protocol needs to be fully specified. The paper should show how KGCL training, validation, hyperparameter selection, downstream classifier training, and test evaluation are separated inside each fold. In particular, because KGCL learns optimized embeddings using labels, the paper must clearly state that label information from the test fold is not used in representation learning. If needed, the authors should add nested cross-validation or re-evaluate the method with a fixed train/validation/test split.

4. The paper must add details of the LLM comparison. Table 5 is used to support a major claim, but the paper does not provide the prompt, number of shots, example selection method, output format, decoding parameters, API version, number of evaluation samples, failure handling, or variance. If the paper compares with LLM-as-a-judge, it should at least show the full protocol on the same test set, sensitivity analysis with multiple prompts, cost and latency measurements, and confidence intervals.

5. The paper should either support the claims about statistical significance with tests or remove those claims. This is especially important for Average F1 on Toxic Comment, where some improvements are very small. For Random Forest, the raw and optimized Avg. F1 are both reported as $0.949$. Without fold-wise paired tests, bootstrap confidence intervals, or effect sizes, statements such as statistically significant improvements across all classifiers are not appropriate.

6. The ablation study needs to be expanded. The current Table 7 only compares parts of contrastive loss and offset loss. The average F1 is $0.78$ for the full model, while several ablated variants have $0.77$. This is not enough to show the necessity of offset loss, the dual-stream design, attention, prototype kernels, orthogonality loss, or magnitude loss. At minimum, the paper should include ablations for removing the semantic basis stream, removing prototype attention, fixed versus learnable prototypes, removing orthogonality loss, removing magnitude loss, fixed versus learned $\alpha$ gate, and different prototype initialization variants.

7. The baselines should be strengthened. The paper compares with raw embeddings, unsupervised SimCSE, standard supervised methods, and few-shot LLMs. However, it lacks comparisons with supervised metric learning or prototype-based baselines that are close to the proposed method. At minimum, the paper should add supervised contrastive learning with a linear head, center loss, triplet or proxy-based metric learning, a simple MLP projection plus classifier, supervised dimensionality reduction instead of only PCA/UMAP, class-balanced classifiers, calibrated logistic regression, and, if possible, a measured comparison with PEFT/LoRA. PEFT is discussed in related work but is not shown as an experimental baseline.

8. The scope of the claims should be limited to the datasets used. GoEmotions is an emotion subset, Amazon Reviews uses star ratings, and Toxic Comment is a toxicity classification task. The paper itself treats GoEmotions and Amazon Reviews as proxies. To generalize these results to broad human value adherence, such as safety, fairness, and helpfulness, the paper needs more direct alignment evaluation datasets or generated-text evaluation datasets. At present, the claim of a general framework for principle evaluation is somewhat too broad.

9. GQI and the geometric analysis should be made more rigorous. The paper should show how each component of GQI is computed, define Class Overlap, report standard deviations and fold-wise variability, and provide results for all datasets. Also, if the paper wants to validate the bound in Theorem 2, it should connect the actual values of $\delta_{\mathrm{intra}}$ and $\delta_{\mathrm{inter}}$ used in the experiments with the observed values of $a(m_i)$ and $b(m_i)$, and report how much the inequalities are satisfied. Table 6 alone is not enough as an experimental validation of the theoretical guarantee.

The following changes are not critical for acceptance, but they would strengthen the paper:

In addition to t-SNE visualizations, the authors should add embedding diagnostics other than UMAP, confusion matrices, per-class precision and recall, calibration, threshold sensitivity, PR-AUC, and qualitative analysis of hard negative examples. For imbalanced data such as Toxic Comment, minority recall, precision, PR-AUC, and the false positive/false negative tradeoff are more important for real deployment than Avg. F1 alone.

---

> ### Author Response · Authors · 2026-06-23
> **Response to Reviewer h6mG (1)**
>
> We sincerely thank the reviewer for the detailed and technically specific review. The comments on final-embedding theory, ordinal modeling, split protocol, LLM reproducibility, statistical uncertainty, ablations, baselines, dataset scope, GQI, and broader impact directly guided the revision. We respond to each requested change below and have updated the paper accordingly (changes are highlighted in red in the revised manuscript).
>
> ### Response to Requested Change 1: Make the theoretical guarantee rigorous for the final embedding
>
> We thank the reviewer for highlighting the distinction between the prototype-mapping space and the final fused normalized embedding.
>
> The intended theoretical contribution is to analyze sufficient conditions for separation in the prototype-mapping stream. The final embedding includes semantic fusion and normalization, so it should be evaluated empirically rather than treated as automatically satisfying the same inequalities.
>
> We have also revised final-embedding wording that could be read as claiming theorem-level final-embedding separation. The manuscript now distinguishes prototype-mapping sufficient conditions from downstream linear-probe evidence: the theory appears in the Method and Appendix `Proofs`, while final embeddings are assessed in the Experiments section through task metrics, paired bootstrap intervals, and GQI diagnostics.
>
> We have revised Theorem 1 and Theorem 2 in the Method section and Appendix B, `Sufficient-Condition Proofs`, to state this boundary explicitly. The revised manuscript explains that final fused embeddings are assessed through downstream metrics, paired bootstrap intervals, and GQI diagnostics. These revisions are shown in the red-highlighted Section 3.4, Appendix B, and the Experiments diagnostics.
>
> ### Response to Requested Change 2: Resolve the ordinal modeling formulation
>
> We thank the reviewer for asking us to make the ordinal formulation precise.
>
> Our intended formulation uses two complementary ordinal biases. Unit-norm angular prototype initialization gives the anchors an ordered starting configuration, while the magnitude loss acts on the pre-normalized prototype mapping `m_i`. Because the final embedding is normalized, ordinal intensity is encouraged during training rather than inferred from the final embedding norm.
>
> We have revised the Method section and Appendix A.2-A.3, `Prototype-Anchor Initialization Details` and `Training and Loss Function Details`, to state this relationship. The method clarification is red-highlighted in the Method section and Appendix A.2-A.3. We also added AmazonReviews ordinal diagnostics in the red-highlighted Table 4: KGCL reduces MAE from raw 0.4784 and SupCon 0.4271 to 0.3039, improves QWK from raw 0.8175 and SupCon 0.8319 to 0.8904, and reduces severe-error rate from raw 0.0637 and SupCon 0.0472 to 0.0205.
>
> ### Response to Requested Change 3: Fully specify cross-validation/split protocol and prevent leakage
>
> We thank the reviewer for emphasizing the importance of split and label-use transparency.
>
> Because KGCL is label-supervised, the revised manuscript makes the data-flow boundary explicit: training labels are used for KGCL training, validation labels are used for checkpoint and representation selection, downstream probes are fit on the training split, and test labels are used only for final metric computation.
>
> We have added a split/protocol table in the Experiments section and a detailed Appendix D.3, `Split Protocol and Leakage Audit`. The verified fixed split sizes are GoEmotions 5126/906/881, AmazonReviews 2271/487/487, and ToxicComment 44256/15073/22609 for train/validation/test. The revised audit states that no test labels are used in representation learning, hyperparameter selection, checkpoint selection, or probe selection. These additions are shown in the red-highlighted Table 1 and Appendix D.3.
>
> ### Response to Requested Change 4: Add full LLM protocol details, sensitivity, cost/latency, and confidence intervals
>
> We thank the reviewer for requesting a fully reproducible LLM comparison.
>
> The revised LLM comparison is presented as a deterministic local-Qwen few-shot diagnostic under a logged protocol. This matches the paper's goal of comparing a lightweight frozen-embedding evaluator against a specified LLM baseline, without making a universal LLM-as-a-judge claim.
>
> We have added the LLM telemetry table in the Experiments section and detailed protocol information in Appendix D.2, `Few-shot LLM Protocol Details`. The appendix reports model, prompt skeleton, shots, demonstration selection, random seed, decoding, parsing, invalid handling, completed calls, token counts, latency, and cost status. We also report prompt-sensitivity diagnostics and paired bootstrap confidence intervals in Appendix D.5 and Appendix D.4. These additions are shown in the red-highlighted Table 10, Appendix D.2, Appendix D.4, and Appendix D.5.

---

> ### Author Response · Authors · 2026-06-23
> **Response to Reviewer h6mG (2)**
>
> ### Response to Requested Change 5: Support or remove statistical significance claims
>
> We thank the reviewer for the suggestion to align claims with uncertainty estimates.
>
> The revised manuscript uses paired bootstrap confidence intervals to distinguish clear differences from small competitive margins. This allows the empirical claims to reflect the strength of the evidence on each dataset.
>
> We have added Table 7, `Paired bootstrap differences in weighted F1`, in the Experiments section. KGCL exceeds the strongest direct baseline on AmazonReviews by 0.0904 weighted F1 with 95% CI [0.0443, 0.1377] and on GoEmotions by 0.0256 with 95% CI [0.0011, 0.0473]. On ToxicComment, the comparison with center loss is a small competitive margin, 0.0009 with 95% CI [-0.0012, 0.0029], and the revised manuscript avoids statistical-superiority wording for that comparison. This addition is shown in the red-highlighted Table 7.
>
> ### Response to Requested Change 6: Expand the ablation study
>
> We thank the reviewer for asking for a more complete ablation analysis.
>
> The revised ablations separate objective-level effects from architecture-level effects. This allows us to test not only whether the full model works, but also how the semantic stream, prototype-attention stream, orthogonality, magnitude, fusion, and prototype choices contribute across tasks.
>
> We have added objective ablations and sensitivity diagnostics in Tables 13 and 14, and architecture diagnostics in Table 15. These results show that the component effects are task-dependent while supporting the revised interpretation of KGCL as a configurable prototype-guided frozen-embedding regularization framework. These additions are shown in the red-highlighted Tables 13-15.
>
> ### Response to Requested Change 7: Strengthen baselines including supervised metric/prototype methods and PEFT
>
> We thank the reviewer for this important baseline suggestion.
>
> The revised manuscript now separates two evaluation settings. The primary setting is frozen-embedding post-hoc regularization, where KGCL is compared against direct metric-learning and prototype baselines under the same frozen-feature and linear-probe protocol. A separate diagnostic table reports supervised adapter, encoder-LoRA, and full fine-tuning results because those methods update additional parameters or the encoder.
>
> We have added SupCon, center loss, triplet loss, prototype classifier, ArcFace, CosFace, frozen residual adapter, encoder-LoRA, and full encoder fine-tuning diagnostics in the Experiments section. This revision makes the baseline comparison stronger while preserving the distinction between frozen-embedding and encoder-updating deployment settings. These additions are shown in the red-highlighted Tables 6 and 8.
>
> ### Response to Requested Change 8: Limit scope to the datasets used
>
> We thank the reviewer for the scope clarification.
>
> The intended empirical claim concerns the three evaluated proxy tasks: selected emotion categorization, ordinal review-intensity prediction, and toxicity classification. These tasks test whether frozen embeddings can be restructured around task-relevant distinctions, but they do not establish broad alignment or human-value evaluation.
>
> We have revised the Abstract, Introduction, Experimental Setup, Conclusion, and Appendix D.7, `Limitations and Scope`, to use "principle-evaluation proxy tasks." The revised manuscript avoids claims about comprehensive alignment, fairness, helpfulness, or generated-text evaluation performance. These scope revisions are shown in red-highlighted statements in the Abstract, Introduction, Experimental Setup, Conclusion, and Appendix D.7.

---

> ### Author Response · Authors · 2026-06-23
> **Response to Reviewer h6mG (3)**
>
> ### Response to Requested Change 9: Make GQI and geometric analysis more rigorous
>
> We thank the reviewer for asking us to make the geometric analysis reproducible.
>
> The intended role of GQI is diagnostic: it summarizes empirical compactness, separation, silhouette behavior, and local class overlap. It is not used as a proof that the theorem's sufficient conditions hold for all final embeddings.
>
> We have added the GQI formula and Class Overlap definition in the Experiments section. Class Overlap is computed as the mean fraction of each sample's 10 nearest neighbors with a different label. We also added all-dataset point estimates in Appendix D.4, Table 23, and state explicitly that GQI does not verify the theorem's inequalities. These additions are shown in the red-highlighted GQI discussion, Table 11, and Appendix D.4 with Table 23.
>
> ### Response to Broader Impact Concern
>
> We thank the reviewer for the broader-impact suggestion.
>
> KGCL is a lightweight post-hoc representation module and should not be interpreted as an autonomous moderation decision-maker. In moderation-like settings, false positives, false negatives, biased labels, and the authority to define evaluated principles all matter.
>
> We have added a Broader Impact discussion in the Conclusion and Appendix D.6, `Broader Impact Statement`. The revised text discusses missed toxic content, over-detection of non-toxic speech, minority dialects, protected-attribute expressions, prototype geometry inheriting training-label bias, human oversight, threshold calibration, auditability, and contestability. These additions are shown in the red-highlighted broader-impact text in the Conclusion and Appendix D.6.